# DaTaSeg: Taming a Universal Multi-Dataset Multi-Task Segmentation Model

**Xiuye Gu**    **Yin Cui**[*][†]    **Jonathan Huang**[*]    **Abdullah Rashwan**[*]    **Xuan Yang**[*]

**Xingyi Zhou**[*]    **Golnaz Ghiasi**    **Weicheng Kuo**    **Huizhong Chen**    **Liang-Chieh Chen**[‡]

**David Ross**
Google Research

## Abstract

Observing the close relationship among panoptic, semantic and instance segmentation tasks, we propose to train a universal multi-**da**taset multi-**ta**sk **seg**mentation model: DaTaSeg. We use a shared representation (mask proposals with class predictions) for all tasks. To tackle task discrepancy, we adopt different merge operations and post-processing for different tasks. We also leverage weak-supervision, allowing our segmentation model to benefit from cheaper bounding box annotations. To share knowledge across datasets, we use text embeddings from the same semantic embedding space as classifiers and share all network parameters among datasets. We train DaTaSeg on ADE semantic, COCO panoptic, and Objects365 detection datasets. DaTaSeg improves performance on *all* datasets, especially small-scale datasets, achieving 54.0 mIoU on ADE semantic and 53.5 PQ on COCO panoptic. DaTaSeg also enables weakly-supervised knowledge transfer on ADE panoptic and Objects365 instance segmentation. Experiments show DaTaSeg scales with the number of training datasets and enables open-vocabulary segmentation through direct transfer. In addition, we annotate an Objects365 instance segmentation set of 1,000 images and release it as a public evaluation benchmark on https://laoreja.github.io/dataseg.

## 1 Introduction

Image segmentation is a core computer vision task with wide applications in photo editing, medical imaging, autonomous driving, and beyond. To suit different needs, various forms of segmentation tasks have arisen, the most popular ones being panoptic [29], semantic [23], and instance [19] segmentation. Prior works generally use specific model architectures tailored to each individual task [28, 45, 21, 6]. However, these segmentation tasks are closely related, as they can all be regarded as grouping pixels and assigning a semantic label to each group. In this paper, we address the following question: *Can we leverage a diverse collection of segmentation datasets to co-train a single model for all segmentation tasks?* A successful solution to this problem would leverage knowledge sharing among datasets, boosting model performance across the board, especially on smaller datasets.

Existing works on unified segmentation models either focus on a single architecture to handle multiple tasks [10, 8, 74, 25], but with separate weights for different datasets; or a single set of weights for

---

[*]Equal contribution. Correspondence to: Xiuye Gu ⟨xiuyegu@google.com⟩.

[†]Work done while at Google. Now at NVIDIA.

[‡]Work done while at Google. Now at ByteDance.

multiple datasets [27, 33, 77], but on the same task. By contrast, in this work, we aim to train a *single* model on *multiple* datasets for *multiple* tasks. Our key idea of unifying these segmentation tasks is to use a universal intermediate mask representation: a set of mask proposals (*i.e.*, grouped pixels) with class labels [63, 10]. Different segmentation tasks can be realized by applying different merge and post-processing operations on this unified representation. This allows us to train our network on the same output space for different tasks. Furthermore, using this representation, we can exploit weak bounding box supervision for segmentation, which is far cheaper to collect than mask annotations.

To encourage knowledge sharing and transfer among the multiple segmentation sources, our network architecture shares the same set of model weights across all datasets and tasks. In addition, we utilize text embeddings as the category classifier, which maps class labels from different datasets into a *shared* semantic embedding space. This design further enhances knowledge sharing among categories with similar meanings in different datasets, *e.g.*, 'windowpane' and 'window-other'. Our approach can be contrasted with an alternative of using dataset-specific model components — we show in our experiments that our simpler, unified approach leads to improved performance and enables open-vocabulary segmentation via simply switching the text embedding classifier.

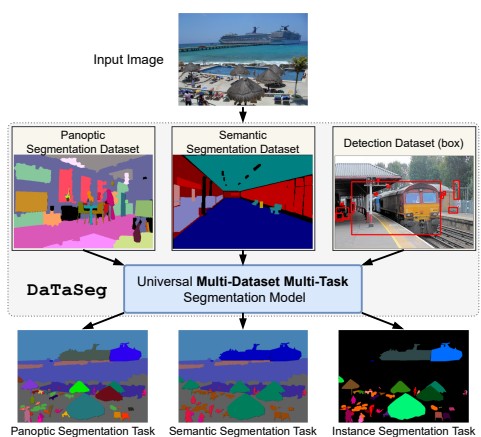

Figure 1: Considering the similarities across various segmentation tasks, and recognizing the potential for enhancing segmentation performance by harnessing data from multiple sources, **we propose to train a universal segmentation model, DaTaSeg, on multiple datasets to perform multiple segmentation tasks.**

Putting these techniques together, we propose **DaTaSeg** as shown in Fig. 1, a universal segmentation model, together with a cotraining recipe for the multi-task and multi-dataset setting. We train DaTaSeg on the ADE20k semantic [75], COCO panoptic [29], and Objects365 detection [56] datasets. We show that DaTaSeg improves performance on all datasets comparing with training separately, and significantly benefits relatively small-scale datasets (ADE20k semantic), outperforming a model trained only on ADE20k semantic by **+6.1** and **+5.1** mIoU with ResNet50 [22] and ViTDet-B [39] backbones. The multi-dataset multi-task setting also allows us to *seamlessly* perform weakly-supervised segmentation by transferring knowledge from other fully supervised source datasets, which we demonstrate on ADE20k panoptic and Objects365 instance segmentation. DaTaSeg also directly transfers to other datasets not seen during training. It outperforms open-vocabulary panoptic segmentation methods on the Cityscapes dataset [11] and performs comparably with open-vocabulary semantic segmentation works on the Pascal Context dataset [49] .

To summarize our contributions, we present DaTaSeg, a single universal segmentation model on multiple segmentation tasks and datasets. DaTaSeg leverages knowledge from multiple sources to boost performance on all datasets. It seamlessly enables weakly-supervised segmentation, directly transfers to other datasets and is capable of open-vocabulary segmentation. As an additional contribution, we have annotated a subset of the Objects365 validation set with groundtruth instance masks and release it as an evaluation benchmark for instance segmentation.

## 2   Related Work

**Muti-dataset training:** Training on multiple datasets has become popular for developing robust computer vision models [64, 78, 33, 47, 61, 67]. For object detection, Wang *et al.* [64] train an object detector on 11 datasets in different domains and show improved robustness. UniDet [78] trains a unified detector on 4 large-scale detection datasets with an automatic label merging algorithm. DetectionHub [47] employs text embeddings to accommodate different vocabularies from multiple datasets, and features dataset-specific designs. For segmentation, MSeg [33] manually merges the vocabularies of 7 semantic segmentation datasets, and trains a unified model on all of them; however, the manual efforts are expensive and hard to scale. LMSeg [77] dynamically aligns segment queries with category embeddings. UniSeg [27] explores the label relation and conflicts between multiple

datasets in a learned way. Most of these works merge datasets using a unified vocabulary on a single task, while our work focuses on a more challenging problem: we merge datasets with different vocabularies *and* different tasks.

**Unified segmentation model:** Panoptic segmentation [29] unifies semantic and instance segmentation. Prior works [28, 66, 69, 7, 52] exploit separate modules for semantic segmentation [45, 5] and instance segmentation [19, 21], followed by another fusion module. Recently, the mask transformer framework proposed by MaX-DeepLab [63] directly predicts masks with class predictions, allowing end-to-end panoptic segmentation. MaskFormer [10] and K-Net [74] adopt a single transformer-based model for different segmentation tasks. Mask2Former [8] improves upon MaskFormer by proposing masked attention, while kMaX-DeepLab [71] develops k-means cross-attention. OneFormer [25] extends Mask2Former with a multi-task train-once design. All these works still train separate weights on *different* datasets. By contrast, we aim at a single unified model that can perform well across multiple segmentation tasks and datasets.

**Weakly-supervised segmentation:** Facing the issue of expensive segmentation mask annotations, many works have proposed to learn segmentation masks from cheaper forms of supervision [50, 12, 42]. In particular, box-supervised approaches are most related to our work, including BoxInst [59], Box2Mask [38], Cut-and-Paste [54] and MAL [34] for instance segmentation, Box2Seg [32] and BCM [58] for semantic segmentation, and DiscoBox [35] and SimpleDoesIt [26] for both semantic and instance segmentation. Despite the popularity of box-driven semantic and instance segmentation, fewer attempts have been made at weakly-supervised panoptic segmentation. Li *et al.* [37] employ a pretrained model to generate pseudo-ground-truth in advance, in addition to weak supervision from bounding boxes and image level labels. Shen *et al.* [57] add a handcrafted branch on top of semantic and instance segmentation models to generate panoptic proposals. Unlike these methods that require customized components, our approach can realize knowledge sharing among datasets with different forms of annotations in a more systematic and data-centric fashion, and seamlessly enable weakly-supervised segmentation.

## 3   Method

### 3.1   A universal segmentation representation

Segmentation aims to group pixels of the same concept together. We propose to use an intermediate and universal representation, namely mask proposals, for all segmentation tasks. A mask proposal is a binary foreground mask with a class prediction. We use this representation for its versatility: one mask proposal can represent a single instance, any number of instances, a region, or even a part of an object. They can overlap and can be combined to form higher-level segmentation outputs — an instance ("thing") or a region of the same semantic class ("stuff"). Thus they are well suited for the different flavors of segmentation considered in this paper. We note that a similar concept has been used in existing works [63, 10, 8, 71], for specific segmentation tasks on a single dataset. We show this representation is especially beneficial in our multi-dataset setting, as different datasets define "thing" and "stuff" categories differently. For example, 'table' is a thing category in ADE20k panoptic dataset, but there is a 'table-merged' stuff category in COCO panoptic. In our framework, both "thing" and "stuff" categories are represented using this same representation, and we treat them differently in the next step.

### 3.2   Merging predictions for specific tasks

**Notations:** We introduce the notations we use in the equations. $\widehat{(\cdot)}$ denotes predictions. Let $(\hat{z}_j, \widehat{M}_j)$ denote the $j$-th mask proposal $\widehat{M}_j$ with its class prediction $\hat{z}_j$, where $\widehat{M}_j \in \mathbb{R}^{H \times W}$ is the (pre-sigmoid) logits for the mask proposal and $\hat{z}_{j,c_k} \in \mathbb{R}$ is the logit for class $c_k$. We assume a fixed set of $N$ mask proposals. $\mathbb{C}_{d,thing}$ and $\mathbb{C}_{d,stuff}$ are the set of thing and stuff categories in dataset $d$.

**Merge operation (MERGE):** Given the varying formats of groundtruth annotations for different segmentation tasks, we propose a MERGE operation to merge the mask proposals predicting the same category $c_k$. We adopt a simple element-wise max operation to merge the mask proposal logits. In particular, the merged mask proposal logits at the $(p, q)$ location for class $c_k$ is computed as:

$$\mathcal{M}(c_k)_{p,q} = \max_j \left( \mathbb{1}[\mathrm{argmax}(\hat{z}_j) = c_k] \cdot \widehat{M}_{j,p,q} \right). \tag{1}$$

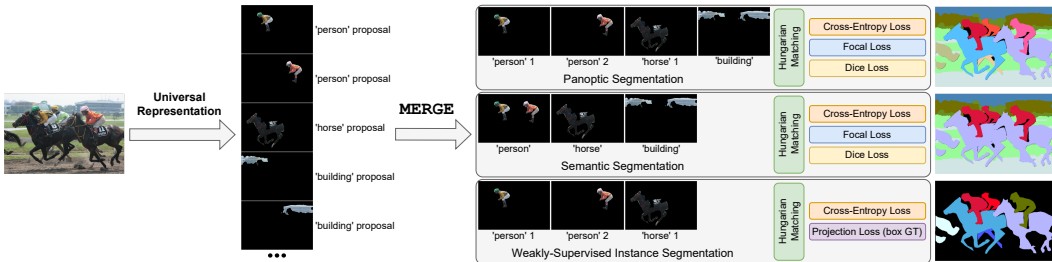

Figure 2: **The universal representation, MERGE operations, and losses for different segmentation tasks.** We use a universal representation for all tasks: a set of mask proposals with class predictions. Then we adopt distinct MERGE operations based on the segmentation task. For panoptic segmentation, we merge proposals predicting the same "stuff" category. For semantic segmentation, both "thing" and "stuff" categories undergo merging. In instance segmentation, we do not perform MERGE and there are no "stuff" categories. During training, we use Hungarian matching, and apply different losses based on the supervision types.

We choose element-wise max so that applying MERGE on raw mask logits is equivalent to applying MERGE on the soft mask predictions post-sigmoid. We also choose to merge at the level of raw logits for numerical stability. If no proposal predicts class $c_k$, then the corresponding merged mask $\mathcal{M}(c_k)_{p,q}$ is set to -100 to ensure it is close to 0 after sigmoid.

In addition to merging masks, we merge their corresponding class predictions by simply averaging the class prediction logits of the proposals predicting class $c_k$:

$$\mathcal{Z}(c_k) = \frac{\sum_j (\mathbb{1}[\arg\max(\hat{z}_j) = c_k] \cdot \hat{z}_j)}{\sum_j (\mathbb{1}[\arg\max(\hat{z}_j) = c_k]) + \epsilon} \, , \qquad (2)$$

where $\epsilon$ is a small number to prevent division by zero.

How we apply the above merge operations depends on the given task, as we now describe:

**Panoptic segmentation:** To make the predicted mask proposals have the same format as the groundtruth, we apply MERGE to all "stuff" categories $c_k \in \mathbb{C}_{d,stuff}$.

**Semantic segmentation:** In contrast with panoptic segmentation, the semantic segmentation groundtruth for each "thing" category is also a single mask (*i.e.*, "thing" categories are treated equivalently as "stuff" categories). We thus apply MERGE to all predicted mask proposals to cover all thing and stuff categories $\mathbb{C}_{d,stuff} \cup \mathbb{C}_{d,thing}$.

**Instance segmentation:** MERGE is not needed in this task, since the desired outputs are separate masks for separate instances. There are no "stuff" categories in the vocabulary ($\mathbb{C}_{d,stuff} = \emptyset$), and therefore no stuff proposals.

**Training:** During training, we apply the corresponding MERGE operations based on the segmentation task. We then use one-to-one Hungarian matching [31] to match the merged outputs with the groundtruth, and calculate the training losses. Detailed training losses are provided in Sec. 3.5.

**Inference:** At inference time, we apply the same MERGE operations to predicted mask proposals based on the given task. We post-process the mask proposals to obtain non-overlapping outputs for panoptic and semantic segmentation. For instance segmentation, we simply take the top mask proposals as final outputs, since overlaps are allowed. Please refer to supplementary for more details.

We illustrate how the MERGE operation works for different segmentation tasks in Fig. 2.

### 3.3   Weakly-supervised instance segmentation

Bounding boxes are much cheaper to annotate than instance masks, and thus detection datasets can have a much larger scale than instance segmentation datasets. In order to train on larger detection datasets with bounding box annotations only, as well as to demonstrate the versatility of our framework to handle various tasks, we propose to perform weak-bounding-box-supervised instance segmentation.

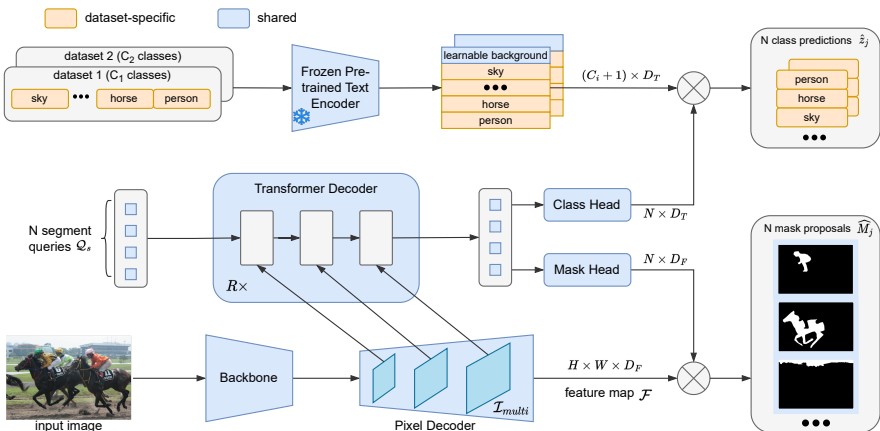

Figure 3: **Overview of our universal multi-dataset multi-task segmentation model (DaTaSeg)**. We feed $N$ learnable segment queries $\mathcal{Q}_s$ into a transformer decoder, which cross-attends to multi-scale image features $\mathcal{I}_{multi}$ from the pixel decoder. The outputs serve as the universal representation for all tasks: $N$ mask proposals $\widehat{M}_j$ accompanied by $N$ class predictions $\hat{z}_j$. To promote knowledge sharing, we share all network parameters across different datasets and tasks. The only dataset-specific design is a classifier consisting of frozen text embeddings of categories specific to each dataset. Additionally, we employ a shared learnable background classifier.

We adopt a simple projection loss $\mathcal{L}_{proj}$ from [60], which measures consistency of vertical and horizontal projections of the predicted masks $\widehat{M}_j$ against groundtruth boxes $b_j$:

$$\mathcal{L}_{proj}(\widehat{M}_j, b_j) = \mathcal{L}_{dice}(\text{Proj}_x(S(\widehat{M}_j)), \text{Proj}_x(b_j)) + \mathcal{L}_{dice}(\text{Proj}_y(S(\widehat{M}_j)), \text{Proj}_y(b_j)), \quad (3)$$

where $\mathcal{L}_{dice}$ is the dice loss [48]; $b_j$ is the groundtruth bounding box matched to $j$-th mask proposal $\widehat{M}_j$; $\text{Proj}_{x/y}$ denotes the projection operation along the $x$ or $y$ axis, which can be implemented by a max operation along the axis; and $S(\cdot)$ denotes the sigmoid function.

By itself, a box consistency loss such as Eqn. 3 is insufficient as a supervision signal for segmentation (e.g., Eqn. 3 is equally satisfied by predicting the bounding box of an object instead of its mask). Thus other works have resorted to additional, often more complex, loss terms (such as the pairwise affinity loss from [60]). However, by training on multiple datasets and multiple segmentation tasks, these handcrafted losses are not necessary, as our model can transfer knowledge gained from other fully-supervised segmentation tasks on other datasets.

### 3.4 Network architecture with knowledge sharing

**Network architecture:** We now describe the network architecture (similar to [8]) that predicts mask proposal and class prediction pairs $(\hat{z}_j, \widehat{M}_j)$. The input image first goes through a *backbone*, and we use a *pixel decoder* to 1) output multi-scale image features $\mathcal{I}_{multi}$, and 2) output a high-resolution feature map $\mathcal{F}$ that fuses information from the multi-scale image features. $N$ mask proposal pairs are then generated from $N$ learnable *segment queries* $\mathcal{Q}_s$: The segment queries are fed into a *transformer decoder*, which cross attends to the multi-scale image features $\mathcal{I}_{multi}$ from the pixel decoder. The $N$ decoder outputs are then passed to class embedding and mask embedding heads (both MLPs) to obtain $N$ (mask embedding, class embedding) pairs. To extract final mask proposals and class predictions, we apply a dot product between the high-resolution feature map $\mathcal{F}$ and the mask embeddings to obtain $N$ mask proposal logits $\widehat{M}_j$. And to obtain class predictions, we compute dot products of the class embeddings against a per-category classifier $w_k$ for each category $c_k$. Fig. 3 shows an overview.

**Knowledge sharing:** We bake knowledge sharing into our model in two ways. First, *all network parameters are shared* across all datasets, allowing knowledge sharing among different datasets and knowledge transfer among different tasks. Additionally, to share knowledge gathered from training samples of similar categories across datasets, *e.g.*, 'windowpane' in ADE20k and 'window-other' in COCO, we propose to *map all category names into a shared semantic embedding space*. We feed the category names into a frozen pre-trained text encoder to set the per-category classifier $w_k$. *i.e.*, we let the fixed text embeddings serve as classifiers. This is similar to open-vocabulary

segmentation [16, 70, 72], but with a different purpose: our focus here is on knowledge sharing among different datasets. This automatic approach scales better than manually unifying label spaces in different datasets [33].

## 3.5 Co-training strategy

We adopt a simple co-training strategy: at each iteration, we randomly sample *one* dataset, then sample the entire batch for that iteration from the selected dataset. This can be contrasted with sampling from multiple datasets in one iteration. The main advantage for our strategy is that it is simple to implement, and allows more freedom to use different settings for different datasets, including applying distinct losses to various tasks. To account for various dataset scales, we use per-dataset sampling ratios, which randomly samples each dataset at a dataset-specific ratio during training, similar to [41].

For fully-supervised tasks, we employ direct *mask supervision*, which can be obtained from panoptic/semantic/instance segmentation groundtruth. To calculate the Hungarian matching cost and the training loss, we use a combination of focal binary cross-entropy loss $\mathcal{L}_{focal}$ [44], and dice loss $\mathcal{L}_{dice}$ [48], following [10]. Regarding weak *bounding box supervision*, we adopt the projection loss $\mathcal{L}_{proj}$ introduced in Sec. 3.3 for the matching cost and training loss. In both cases of supervision, we use the negative of the class prediction probability $p$ as the classification matching cost [4], and use cross-entropy loss $\mathcal{L}_{ce}$ for classification training. The total training loss is:

$$\mathcal{L}_d = \lambda_{ce,d}\mathcal{L}_{ce} + \lambda_{focal,d}\mathcal{L}_{focal} + \lambda_{dice,d}\mathcal{L}_{dice} + \lambda_{proj,d}\mathcal{L}_{proj}. \tag{4}$$

The Hungarian matching cost is defined similarly:

$$\mathcal{C}_d = -\mu_{ce,d} \cdot p + \mu_{focal,d}\mathcal{L}_{focal} + \lambda_{dice,d}\mathcal{L}_{dice} + \mu_{proj,d}\mathcal{L}_{proj}. \tag{5}$$

Here, $\lambda_{*,d}$ and $\mu_{*,d}$ are the weights for dataset $d$.

## 4 Experiments

**Datasets and metrics:** We train and evaluate DaTaSeg on COCO panoptic [29] and ADE20k semantic [75] using mask supervision, as well as Objects365-v2 [56] detection datasets using bounding box weak supervision. **COCO panoptic** is the most popular panoptic segmentation benchmark with 118,287 training images and 5,000 validation images. COCO has 80 thing categories and 53 stuff categories. **ADE20k semantic** is one of the most widely used semantic segmentation benchmarks with 150 categories, 20,210 training images, and 2,000 validation images.

For evaluation, besides the training datasets, we also evaluate on **ADE20k panoptic**, which uses the same validation images as ADE20k semantic but with panoptic annotations. The original 150 categories are divided into 100 thing categories and 50 stuff categories. Finally, we train on the **Objects365-v2 detection** (bounding boxes only) dataset, which has 365 categories and 1,662,292 training images. To evaluate the weakly-supervised instance segmentation results, we manually label a subset of 1,000 images from the Objects365 validation set. Our annotation pipeline involves 20 well-trained human annotators, following the protocol in [2], without using any automatic assistant tools. Please refer to Appendix B for more details. We release this **Objects365 instance segmentation evaluation set** as a public benchmark.

To compare DaTaSeg with the state-of-the-art methods, we in addition evaluate on popular open-vocabulary segmentation benchmarks. Following OpenSeg [16], we evaluate on **PASCAL Context** datasets [49] with 5k val images. We use its two versions: 59 classes (**PC-59**) and 459 classes (**PC-459**). We also evaluate on the 500-image val set of **Cityscapes** panoptic dataset [11], which focuses on urban street scenes with 11 stuff categories and 8 thing categories.

We report results in standard evaluation metrics: panoptic quality (**PQ**), mean intersection-over-union (**mIoU**), and mean average precision (**AP**) for panoptic, semantic, and instance segmentation, respectively.

**Implementation details:** We experiment with ResNet [22] and ViTDet [39] backbones. For ResNet, we use an ImageNet-1K [55] pretrained checkpoint, with a backbone learning rate multiplier of 0.1 [4]; the pixel decoder consists of a transformer encoder [62] and a modified FPN [43] following [10],

| Backbone | Model | Fully-Supervised | | Weakly-Supervised Transfer | |
|---|---|---|---|---|---|
| | | ADE semantic mIoU | COCO panoptic PQ | ADE semantic → panoptic PQ | O365 box → instance mask AP |
| ResNet50 | Separate | 42.0 | 48.2 | 26.9 | 12.3 |
| | DaTaSeg | 48.1 (+6.1) | 49.0 (+0.8) | 29.8 (+2.9) | 14.3 (+2.0) |
| ViTDet-B | Separate | 46.3 | 51.9 | 27.5 | 14.7 |
| | DaTaSeg | 51.4 (+5.1) | 52.8 (+0.9) | 32.9 (+5.4) | 16.1 (+1.4) |
| ViTDet-L | DaTaSeg | 54.0 | 53.5 | 33.4 | 16.4 |

Table 1: **Comparing DaTaSeg with separate dataset-specific models.** We show results on both the training tasks (fully-supervised) and new tasks (weakly-supervised transfer). Our single DaTaSeg outperforms separately trained models on *all* datasets. They are trained under the same settings, so the performance gains come from knowledge in other datasets through our multi-dataset multi-task cotraining recipe. We observe: 1) DaTaSeg *significantly* improves tasks with limited data (ADE20k semantic); 2) DaTaSeg enables weakly-supervised knowledge transfer (ADE20k panoptic and O365 instance); 3) DaTaSeg scales well with backbones.

| Training sets | | | Fully-Supervised | | Weakly-Supervised Transfer | |
|---|---|---|---|---|---|---|
| ADE semantic | COCO panoptic | O365 bbox | ADE semantic mIoU | COCO panoptic PQ | ADE semantic → panoptic PQ | O365 box → instance mask AP |
| ✓ | | | 42.0 | 8.4 | 26.7 | 1.0 |
| | ✓ | | 15.3 | 48.2 | 11.6 | 5.8 |
| | | ✓ | 11.0 | 12.4 | 5.8 | 12.3 |
| ✓ | | ✓ | 46.3 (+4.3) | 14.0 | 26.4 (-0.3) | 15.0 (+2.7) |
| | ✓ | ✓ | 18.3 | 48.9 (+0.7) | 12.3 | 15.2 (+2.9) |
| ✓ | ✓ | | 47.3 (+5.3) | 49.0 (+0.8) | 30.5 (+3.8) | 5.6 |
| ✓ | ✓ | ✓ | 48.1 (+6.1) | 49.0 (+0.8) | 29.8 (+2.9) | 14.3 (+2.0) |

Table 2: **Importance of training on multiple datasets.** We train DaTaSeg on all combinations the three datasets. Experiments are done on ResNet50 under same settings. DaTaSeg scales with the number of training datasets.

except that we use Layer Norm [1] and GeLU [24] activation for training stability. For ViTDet, we use the ImageNet-1K MAE pretrained checkpoint [20] with the layerwise lr decay [39]; the pixel decoder is the simple feature pyramid inside ViTDet, whose outputs are upsampled and added together to get the high resolution feature map $\mathcal{F}$. The mask embedding head is a 3-layer MLP. The class embedding head contains a linear layer followed by ReLU. We use CLIP-L/14 [53] as the pretrained text encoder. We use 100 segment queries $\mathcal{Q}_s$, unless otherwise stated. See Appendix H for more details.

## 4.1 DaTaSeg improves over dataset-specific models

As an apples-to-apples comparison, we separately train a model on each dataset. We ensure that the models see the same number of images in cotraining and separately-training on each dataset. Table 1 (left) shows the results. DaTaSeg outperforms separately trained models on all datasets. Since we use exactly the same settings, we attribute the performance boost to our multi-dataset multi-task training recipe, which harnesses knowledge from multiple sources. Especially, DaTaSeg leads to a significant performance boost on ADE20k semantic: **+6.1 mIoU** and **+5.1 mIoU** with ResNet50 and ViTDet-B backbones, respectively. This proves our argument that cotraining on more data helps overcome the data limitation issue in smaller-scale segmentation datasets.

## 4.2 DaTaSeg enables weakly-supervised transfer

In Table 1 (right), we evaluate DaTaSeg on ADE20k panoptic and Objects365 instance segmentation tasks. Note we only have weak supervision (ADE20k semantic and Objects365 bbox) during training. Again, we compare to models separately trained on one dataset as our baselines. Specifically, we use the ADE semantic for ADE panoptic evaluation, and an Objects365 model trained only using weak box supervision for Objects365 instance evaluation, respectively.

By cotraining DaTaSeg in a multi-dataset multi-task fashion, we improve the ADE20k panoptic performance by **+2.9 PQ** and **+5.4 PQ** with ResNet50 and ViTDet-B backbones. Our design allows the panoptic segmentation knowledge to transfer from COCO to ADE20k. For reference, the fully-supervised performance is 37.7 PQ on ResNet50 and 42.3 PQ on ViTDet-B.

| Method | Backbone | Training data | PC-59 mIoU | PC-459 mIoU | COCO mIoU | Cityscapes PQ |
|---|---|---|---|---|---|---|
| ODISE [68] | UNet+M2F | LAION+CLIP+COCO | 55.3 | 13.8 | 52.4 | 23.9 |
| MaskCLIP [13] | R-50 | COCO pan+CLIP | 45.9 | 10.0 | – | – |
| OVSeg [40] | R-101c | COCO stuff+cap | 53.3 | 11.0 | – | – |
| | Swin-B | | 55.7 | 12.4 | – | – |
| OpenSeg [16] | R-101 | COCO pan+cap | 42.1 | 9.0 | 36.1 | – |
| | Eff-b7 | COCO+Loc. Narr. | 44.8 | 11.5 | 38.0 | – |
| DaTaSeg | R-50 | COCO panoptic | 50.9 | 11.1 | 57.7 | 30.0 |
| | ViTDet-B | +ADE semantic | 51.1 | 11.6 | 62.7 | 28.0 |
| | ViTDet-L | +O365 bbox | 51.4 | 11.1 | 62.9 | 29.8 |

Table 3: **Compare DaTaSeg to state-of-the-art open-vocabulary segmentation models.** We apply DaTaSeg directly to other semantic or panoptic segmentation datasets without finetuning. We compare to ODISE [68], MaskCLIP [13], OVSeg [40], and OpenSeg [16] on their corresponding benchmarks. We only conduct system-level comparison due to differences in training data and backbones. DaTaSeg performs comparably on all semantic segmentation benchmarks and outperforms the recent work ODISE on Cityscapes. PC: Pascal Context. COCO: COCO is included in the training set for all models, and is not completely "open-vocabuarly".

| # queries | | 50 | 100 | 150 |
|---|---|---|---|---|
| ADE semantic | mIoU | 47.4 | 48.1 | 48.7 |
| COCO panoptic | PQ | 48.7 | 49.0 | 48.9 |
| ADE panoptic[†] | PQ | 30.7 | 29.8 | 29.1 |
| O365 instance[†] | AP | 13.4 | 14.3 | 15.8 |

Table 4: **In general, the performance of DaTaSeg increases as we increase the number of segment queries.** Besides, using only 50 queries already achieves good performance. Experiments are conducted using a ResNet50 backbone. [†]: weakly-supervised tasks.

| | | DaTaSeg | +D-S modules |
|---|---|---|---|
| ADE semantic | mIoU | 48.1 | 48.1 (-0.0) |
| COCO panoptic | PQ | 49.0 | 46.0 (-3.0) |
| ADE panoptic[†] | PQ | 29.8 | 26.9 (-2.9) |
| O365 instance[†] | AP | 14.3 | 10.9 (-3.4) |

Table 5: **Adding dataset-specific (D-S) modules hurts performance on almost all datasets.** This shows the importance of sharing all parameters for better knowledge sharing, which particularly benefits the weakly-supervised tasks. Experiments are conducted on a ResNet50 backbone.

On Objects365, we only train with a weak box consistency loss $\mathcal{L}_{proj}$. The multi-dataset multi-task training recipe improves the mask AP by **+2.0 AP** and **+1.4 AP** for the two backbones. The improvement is most likely from the instance segmentation knowledge in COCO panoptic.

### 4.3 DaTaSeg scales with size of backbones

Our DaTaSeg design is orthogonal to detailed network architectures, as long as it is able to output the universal representation (mask proposals with class predictions). This is supported by the consistent performance gains among ResNet50, ViTDet-B, and ViTDet-L backbones as shown in Table 1. It also scales well as we increase the sizes of the backbones. With ViTDet-L, DaTaSeg reaches **54.0 mIoU** on ADE20k semantic and **53.5 PQ** on COCO panoptic in a single model, without bells and whistles.

### 4.4 DaTaSeg scales with number of datasets

We study how DaTaSeg scales with the number of training datasets. We train DaTaSeg on all combinations of one, two, and three datasets among the ADE20k semantic, COCO panoptic, and Objects365 detection datasets, in order to conduct a comprehensive study, with a ResNet50 backbone. Table 2 presents the results. Looking at each column, we see that performance on each dataset generally improves as the number of training datasets increases, especially for ADE semantic.

Since DaTaSeg shares all parameters among all datasets and tasks, we can evaluate the cross-dataset transfer performance. We notice the model transfers to datasets that are not trained. *e.g.*, A model trained on COCO panoptic and Objects365 detection achieves **18.3 mIoU** on ADE semantic, which is comparable to open-vocabulary segmentation performance (LSeg+ [16, 36] achieves 18.0 mIoU on a ResNet101 backbone).

### 4.5 DaTaSeg enables open-vocabulary segmentation

A further advantage of our fully-shared architecture is that we have the ability to directly transfer to other segmentation datasets. We simply switch the text embedding classifier with the vocabularies

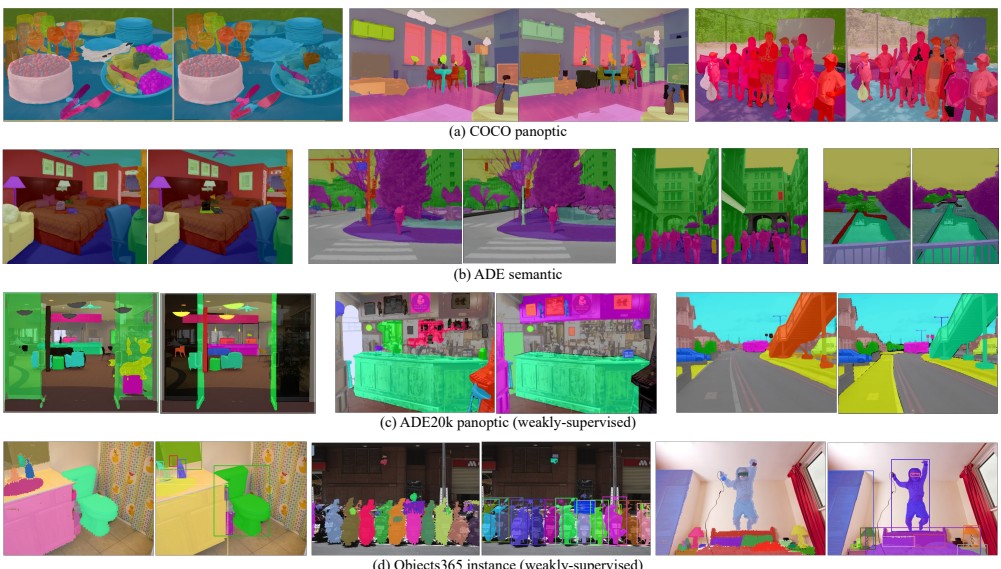

(a) COCO panoptic

(b) ADE semantic

(c) ADE20k panoptic (weakly-supervised)

(d) Objects365 instance (weakly-supervised)

Figure 4: **Qualitative results of DaTaSeg on all datasets & tasks.** For every pair of images, the left is DaTaSeg prediction and the right is groundtruth. DaTaSeg succeeds on hard cases (*e.g.*, transparent wine glasses on the top left) as well as weakly-supervised datasets. For object instances in panoptic and instance segmentation, the colors are not matched to the groundtruths. Black denotes ignored regions.

in a target dataset not used in training. Table 3 shows the results comparing DaTaSeg to several open-vocabulary works: ODISE [68] and MaskCLIP [13] are open-vocabulary panoptic segmentation methods and OVSeg [40] and OpenSeg [16] are open-vocabulary semantic segmentation approaches. Unlike these works, our method does not train on large-scale image-text data or use pretrained image-text models (except the pretrained text encoder), which puts our method at a disadvantage. Despite this disadvantage, DaTaSeg outperforms ODISE on the Cityscapes panoptic dataset. Note that there is a domain gap between our training sets and Cityscapes, which focuses on urban street scenes. DaTaSeg achieves comparable performance on PC-59 and PC-459. All methods in comparison train on COCO panoptic and ours has the best performance on COCO semantic. These results show that DaTaSeg enables open-vocabulary segmentation via our multi-dataset multi-task training approach.

## 4.6 DaTaSeg scales with number of queries

We perform ablation study on the number of segment queries $\mathcal{Q}_s$ and show the results in Table 4. The performance improves as we increase the number of queries from 50 to 150, on almost all datasets. Besides, DaTaSeg achieves reasonably good performance with only 50 queries, which may benefit memory-limited application scenarios. We perform more ablation studies in Appendix D.

## 4.7 Adding dataset-specific modules hurts performance

In multi-dataset learning, prior works have introduced dataset-specific modules [47], in order to address inconsistencies across datasets [33, 61]. However, such a design may weaken knowledge sharing across different datasets and tasks, especially for weakly-supervised tasks that may rely more on other datasets and tasks to compensate for the weak supervision. To study this issue thoroughly, we carefully design several dataset-specific modules, including dataset-specific queries, dataset-specific heads, and dataset-specific prompts. These modules are lightweight by design so as to still encourage knowledge sharing through other shared parameters.

In Table 5, we evaluate the effectiveness of these dataset-specific modules by adding them to DaTaSeg, and keeping all other settings the same. Comparing with DaTaSeg that shares all parameters, results reveal that using dataset-specific modules hurts performance on all datasets except ADE20k semantic. Moreover, we see that the performance drops more on weakly-supervised tasks (ADE panoptic and O365 instance). This verifies that removing dataset-specific modules yields better knowledge sharing across datasets and tasks, and thus greatly improves performance. See Appendix E for more details.

## 4.8 Qualitative analysis

We show qualitative results (with ViTDet-L) in Fig. 4. DaTaSeg performs well in both fully and weakly supervised settings. We observe that localization quality is good in general, while classification is more error-prone. Nevertheless, DaTaSeg succeeds in challenging cases including transparent objects, occlusion, and crowd scenes.

## 5 Conclusion

The goal of our work is to train a single universal segmentation model on multiple datasets and multiple tasks (*i.e.*, semantic, instance and panoptic segmentation). We present DaTaSeg, which leverages a universal segmentation representation, shared network parameters, and shared semantic embedding space for classification. DaTaSeg surpasses separately trained models, leading to significant gains on smaller datasets such as ADE20k semantic. It unlocks a new weakly-supervised transfer capability on datasets that were not explicitly trained for a particular task (*e.g.*, ADE20k panoptic and Objects365 instance). DaTaSeg also directly transfer to more segmentation datasets and enables open-vocabulary segmentation. We believe that in the future, this multi-task multi-dataset setting will be the norm rather than the outlier as it currently is, and we see our paper as taking a step in this direction.

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

## Appendix

Funding in direct support of this work: Google LLC. There is no additional revenues related to this work.

## A  Broader Impacts

We propose a single framework for multi-dataset multi-task segmentation, which especially benefits memory-limited application scenarios, *e.g.*, our work can make a single segmentation model that meets all kinds of segmentation needs on memory-limited mobile devices (semantic segmentation for photo editing, instance segmentation for object detection and image search, *etc.*), instead of using separate models for various segmentation tasks.

Moreover, our approach improves segmentation performance from a data-centric view: We leverage segmentation data from multiple sources to improve the overall segmentation performance, especially on datasets of smaller scales, *e.g.*, ADE20k semantic. We also transfer knowledge from other sources to enable weakly-supervised segmentation. This is helpful on the scenario where segmentation data are very hard to collect, we can cotrain on similar data to help improve the performance, or even train on data with weaker supervision. *e.g.*, if we want to train a segmentation model on rare wild animals, we can co-train a model on a small set of these kinds of animals, and a larger dataset of more common animals.

As for bias and fairness, our model learns from the training data, so it may inherit the bias and fairness issues existing in the training data. Nonetheless, our model trains on multiple sources, and the fairer datasets may compensate the bias and fairness issues in other datasets.

## B  More about our newly labeled Objects365 instance segmentation evaluation dataset

We randomly sampled 1,000 images from the Objects365 V2 validation set. Different from the "federated" LVIS dataset [18] (not all categories in each image are completely labeled), the bounding boxes in Objects365 are completely labeled for all categories and we follow this complete annotation in our instance masks. For each bounding box annotation, we generate a foreground/background annotation question for the raters. In the question, raters are asked to paint a semi-transparent instance mask on the original image, with the groundtruth bbox shown on the image as a guide and the groundtruth category displayed on the side. We do not crop the original image to the bbox region, but show the entire image to provide more context information, which is especially helpful for small objects. If the boundary is too blurry or too dark to annotate the instance mask, the raters can skip the question.

The annotation tool is a free painting tool, which allows the raters to freely draw the instance mask. We ask the raters to try to draw within the bbox, but if the object is obviously exceeding the bbox, then they can draw outside the bbox. The size of the stroke is adjustable. Raters can zoom-in/out, draw straight lines, and fill the inside of a plotted boundary with a single click (which is quite useful for instance segmentation). Unlike polygons in COCO *etc.*, the annotation tool allows us to save higher-resolution binary masks.

We inserted a total of 13,372 questions. Following the common practice in instance segmentation, we did not label the crowd instances, as they are skipped in both training and inference. In the end, we obtained 12,836 valid instance mask annotations. It took a total of 800.59 rater hours. On average, raters spent 3.74 minutes on each valid mask.

In Fig. 5, we show some statistics about our Objects365 instance segmentation dataset. The number of masks per category follows a relatively long-tailed distribution (Fig. 5(a)). The number of masks per image reveals that a large number of images have many instance annotations (Fig. 5(b)). And the statistics of the relative mask size is close to that of the LVIS datasets (Fig. 5(c)).

As visualized in Fig. 6, the mask annotations are very high-quality. The raters carefully handled small objects, thin structures, complicated shapes, occluded objects, *etc.*

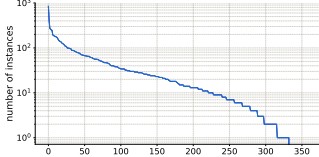 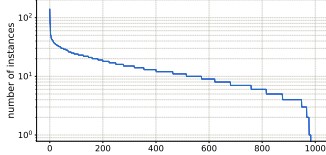 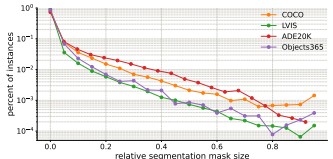

(a) Number of annotated instance masks per category. It follows a relatively long tail distribution.

(b) Number of annotated instance masks per image. More than 500 images have more than 10 instances.

(c) Relative segmentation mask size (square root of the ratio of mask area to image area), compared with COCO, LVIS, and ADE20K. Our statistics are similar to LVIS.

Figure 5: **Dataset statistics for our Objects365 instance segmentation evaluation set.**

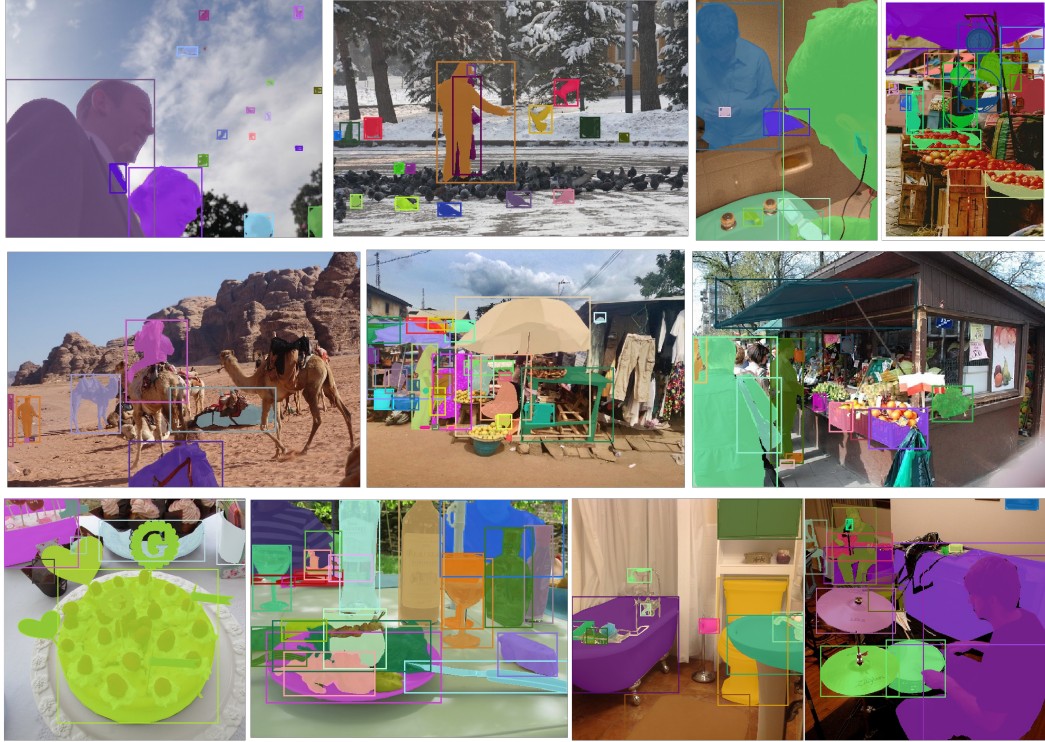

Figure 6: **Visualization of our Objects365 instance segmentation dataset.** We store binary masks, which has higher mask quality compared with polygons. The quality on small objects are good (first two images). Complex structures are also labeled delicately, *e.g.*, the camels (1st image on 2nd row) and the decoration on the cake (1st image on 3rd row). For occlusions, raters carefully avoided the occluded regions as they are not a part of the object to annotate, *e.g.*, the fruits in the basket are not included in the basket masks (last image on 1st row).

For the objects which do not have an instance mask since raters skipped it, it's possible that the model still predicts an instance mask for that object. We include the skipped annotations to make it optional to ignore them during evaluation, so as to avoid treating such cases as false positive.

We release this dataset. We believe this effort can provide the research community a good benchmark for instance segmentation evaluation, *e.g.*, for evaluating weakly-supervised instance segmentation learned from Objects365 bounding boxes.

## C   Open-vocabulary semantic segmentation on PASCAL VOC

We evaluate DaTaSeg on the PASCAL VOC 2012 semantic segmentation dataset [14], which includes 20 object classes and a background class. We directly transfer our trained model to its validation set with 1,449 images without finetuning. We follow the setting in MaskCLIP+ [76] and ignore the background class during evaluation.

| Method | Backbone | mIoU | mIoU-unseen | mIoU-seen |
|---|---|---|---|---|
| SPNet [65] | | 56.9 | 0.0 | 75.8 |
| SPNet-C [65] | | 63.2 | 15.6 | 78.0 |
| ZS3Net [3] | | 61.6 | 17.7 | 77.3 |
| CaGNet [17] | DeepLabv2-ResNet101 | 65.5 | 26.6 | 78.4 |
| STRICT [51] | | 70.9 | 35.6 | 82.7 |
| MaskCLIP+ [76] | | 88.1 | 86.1 | 88.8 |
| Fully-supervised | | 88.2 | 87.0 | 88.6 |
| | ResNet-50 | 89.4 | 87.7 | 89.9 |
| DaTaSeg | ViTDet-B | 92.8 | 90.9 | 93.4 |
| | ViTDet-L | 94.7 | 94.3 | 94.9 |

Table 6: **Open-vocabulary semantic segmentation on the PASCAL VOC 2012 dataset.** DaTaSeg outperforms all other zero-shot/open-vocabulary methods, and even the fully-supervised baseline. Comparison methods are trained on 15 seen classes of PASCAL VOC and only 5 classes are unseen (mIoU-unseen). In contrast, DaTaSeg has never seen the PASCAL VOC dataset during training (it's trained on several other datasets). We attribute DaTaSeg's performance improvement to our multi-dataset multi-task training. Following [76], the background class is ignored. The numbers of the comparison methods are from [76]. SPNet-C stands for SPNet with calibration.

| | Fully-supervised | | Weakly-supervised transfer | |
|---|---|---|---|---|
| Dataset sampling ratio | ADE semantic mIoU | COCO panoptic PQ | ADE semantic → panoptic PQ | O365 box → instance mask AP |
| 1:4:4 | 48.1 | 49.0 | 29.8 | 14.3 |
| 1:2:2 | 46.8 | 48.6 | 29.1 | 12.8 |
| 1:1:1 | 45.3 | 48.0 | 28.4 | 13.7 |

Table 7: **Ablation study on dataset sampling ratio (ADE:COCO:O365).** Evaluated on a ResNet50 backbone. Results show that our adopted sampling ratio (1:4:4) is better than the other sampling ratios.

We compare with other zero-shot and open-vocabulary segmentation methods in Table 6, which are trained on 15 seen classes with 5 classes held out during training as unseen classes (potted plant, sheep, sofa, train, and TV monitor). DaTaSeg has never seen the PASCAL VOC dataset during training, though it is trained on other segmentation datasets (COCO panoptic, ADE semantic, and Objects365 detection). DaTaSeg with a ResNet-50 backbone outperforms all these methods with DeepLabv2-ResNet101 backbones. Surprisingly, DaTaSeg even outperforms the fully-supervised counterpart. We attribute DaTaSeg's performance improvement to our multi-dataset multi-task training. We note this is not an apple-to-apple comparison, since the training sources are different. The main purpose of this comparison is to show how our "data-centric" DaTaSeg is positioned against zero-shot/open-vocabulary segmentation methods on the PASCAL VOC dataset, using a direct transfer manner.

# D   Ablation studies

We perform more ablation studies in this section.

**Ablation study on dataset sampling ratio:** Our dataset sampling strategy (Sec. 3.5) avoids the imbalance of dataset sampling by specifying the per-dataset sampling ratio. The proportion of training samples coming from each dataset in the whole training process is determined by that sampling ratio. In our main results, the sampling ratio is 1:4:4 for ADE:COCO:O365. We ablate the dataset sampling ratio in Table 7. Results show that our adopted sampling ratio is better than the other sampling ratios.

**Ablation study on the portion of Objects365 training data:** Readers may have the question about the trade-offs between the quality and quantity of the training samples when we introduce the weakly-supervised training data (Objects365). In Table 2, we show the results with and without cotraining on Objects365. The performance is generally better when cotraining on Objects365. In addition, we experiment with cotraining on different portions of the dataset: we cotrain DaTaSeg on the full COCO panoptic and ADE semantic datasets, and 10% / 25% / 50% / 100% of O365 training data. We show the results in Table 8. Results show that when increasing the number of O365 weakly-supervised training samples, O365 performance increases, COCO panoptic performance is not affected, and ADE20k performance slightly decreases. Overall, the gains are larger than the losses.

| Portion of O365 training data | Fully-supervised | | Weakly-supervised transfer | |
|---|---|---|---|---|
| | ADE semantic mIoU | COCO panoptic PQ | ADE semantic → panoptic PQ | O365 box → instance mask AP |
| 10% | 48.5 | 48.7 | 30.0 | 10.4 |
| 25% | 48.6 | 48.3 | 30.6 | 12.0 |
| 50% | 47.7 | 48.5 | 29.8 | 13.1 |
| 100% | 47.2 | 48.7 | 29.4 | 14.5 |

Table 8: **Ablation study on the portion of O365 training data.** Evaluated on a ResNet50 backbone. When increasing the number of O365 weakly-supervised samples, O365 performance increases, COCO panoptic performance is not affected, and ADE20k performance slightly decreases.

| $\lambda_{proj}$ | $\mu_{proj}$ | ADE semantic | COCO panoptic | ADE semantic → panoptic | O365 box → instance |
|---|---|---|---|---|---|
| 5.0 | 1.0 | 45.5 | 48.3 | 28.4 | 12.5 |
| 2.0 | 0.5 | 47.2 | 48.7 | 29.4 | 14.5 |

Table 9: **Ablation study on the projection loss $\mathcal{L}_{proj}$ weights.** Evaluated on a ResNet50 backbone. The results indicate that the 2nd setting (our final setting) has a better performance on all datasets.

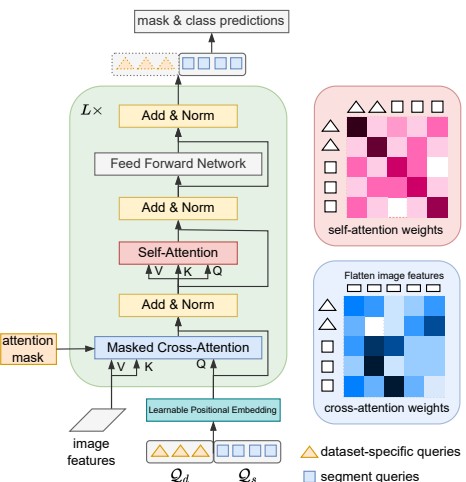

Figure 7: **Dataset-specific queries in the transformer decoder.** We use different dataset-specific queries $\mathcal{Q}_d$ for each dataset $d$, which are treated the same as the segment queries $\mathcal{Q}_s$, and involved in both cross-attention and self-attention. All dataset-specific queries are discarded before predicting masks and classes.

**Ablation study on projection loss $\mathcal{L}_{proj}$ weights:** Our settings of $\lambda$ and $\mu$ (Table 13) closely follow Maskformer series [10, 8]. We add the the bounding-box projection loss $\mathcal{L}_{proj}$ for Objects365. We run an ablation study on the weights ($\lambda_{proj}$ and $\mu_{proj}$) for the projection loss in Table 9. The results indicate that the 2nd setting (our final setting) has a better performance on all datasets. Finding the optimal weights for multiple losses is an interesting yet challenging problem, and we leave it for future work.

# E   Adding dataset-specific modules hurts performance

In this section, we first introduce our designed dataset-specific modules and then show more ablation studies on these modules.

As prior works have pointed out [33, 61], there are many inconsistencies across segmentation datasets, including labels having their own particular definition, being annotated using different protocols, *etc*. We explore if adding dataset-specific modules can improve overall performance by designing the following modules.

**Dataset-specific queries:** In our architecture design introduced in Sec. 3.4, an input image will not get any dataset-specific information until the text embedding classifier, which is almost at the final stage of the network. We thus propose to add a few dataset-specific queries $\mathcal{Q}_d$ as an auxiliary input to the beginning of transformer decoder. Each training dataset $d$ has its own $\mathcal{Q}_d$. The dataset-specific

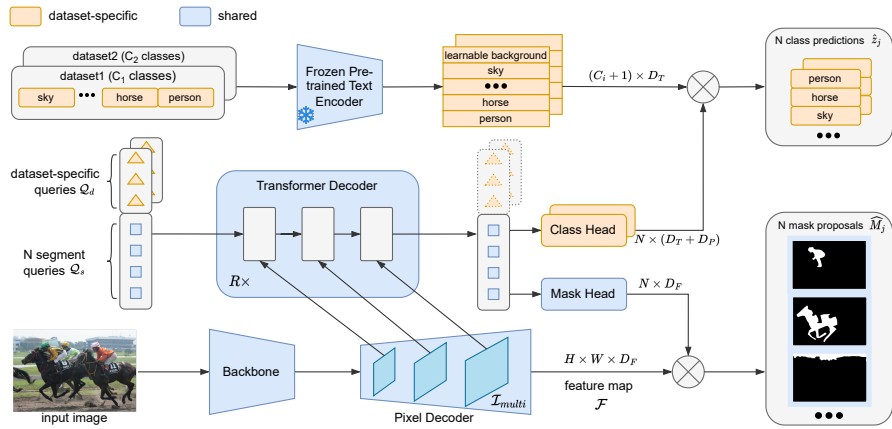

Figure 8: **Overview of incorporating all our designed dataset-specific modules into our universal multi-dataset multi-task segmentation model (DaTaSeg)**. We design a set of *dataset-specific queries* fed into the transformer decoder, *dataset-specific classification head*, and *dataset-specific background classifier*. They are highlighted in orange. All these modules are learnable. Different datasets have separate parameters for these modules.

queries are treated the same as regular segment queries $\mathcal{Q}_s$. The only difference is that we do not make mask and class predictions for the dataset-specific queries. Fig. 7 shows the details. In the cross-attention layer, $\mathcal{Q}_d$ cross attends to the image features — the cross-attention weight is:

$$\mathcal{W}_{\text{cross\_atten}} = \text{softmax}(\mathcal{A} + W_q [\mathcal{Q}_d \ \mathcal{Q}_s] \cdot \mathcal{I}_{multi}^T W_k^T), \tag{6}$$

where $\mathcal{I}_{multi}$ are the multi-scale image features in Sec. 3.4. $W_q, W_k$ denote the linear projection weights; $\mathcal{A}$ denotes the attention mask.

In the self-attention layers, there are interactions for all pairs of $\mathcal{Q}_d$ and $\mathcal{Q}_s$, allowing the segment queries to be aware of the dataset-specific information at the beginning of transformer decoder. This can be shown by the self-attention weight calculation:

$$\mathcal{W}_{\text{self\_atten}} = \text{softmax}(W_q \begin{bmatrix} \mathcal{Q}_d \\ \mathcal{Q}_s \end{bmatrix} \begin{bmatrix} \mathcal{Q}_d^T & \mathcal{Q}_s^T \end{bmatrix} W_k^T)$$

$$= \text{softmax}(W_q \begin{bmatrix} \mathcal{Q}_d \mathcal{Q}_d^T & \mathcal{Q}_d \mathcal{Q}_s^T \\ \mathcal{Q}_s \mathcal{Q}_d^T & \mathcal{Q}_s \mathcal{Q}_s^T \end{bmatrix} W_k^T). \tag{7}$$

**Dataset-specific classification head and background classifier:** We observe that dataset inconsistency tends to primarily be a classification issue, rather than localization issue. Hence, for all datasets we use the same set of parameters for localization (backbone, pixel decoder, transformer decoder, mask embedding head) to enhance knowledge sharing. For classification, we design some dataset-specific components: specifically, we use a dataset-specific MLP class embedding head as well as a learnable dataset-specific background classifier, since the definitions of background vary from dataset to dataset.

**Overall dataset-specific architecture:** The overall architecture equipped with all our dataset-specific modules is shown in Fig. 8. We design the dataset-specific modules to be light-weight, which allows us to save on memory costs. With this design of a mixture of a mostly shared network and light-weight dataset-specific modules, the model has the freedom to choose whether to leverage dataset-specific information, or to use the shared knowledge across datasets/tasks. However, one significant disadvantage of adding dataset-specific modules is that it's hard to decide which set of dataset-specific parameters to use when directly transferring to other datasets (Sec. 4.5), and thus it *hurts the open-vocabulary capability*.

**Ablation studies on dataset-specific modules:** We ablate on the number of dataset-specific queries $\mathcal{Q}_d$ in Table 10. It shows that removing the dataset-specific queries achieves the best results on COCO panoptic and Objects365 by a large margin. It also achieves reasonable performance on other datasets. Using 30 queries hurts the performance on COCO panoptic and ADE semantic.

| # of dataset-specific queries | Fully-supervised | | Weakly-supervised transfer | |
|---|---|---|---|---|
| | ADE semantic mIoU | COCO panoptic PQ | ADE semantic → panoptic PQ | O365 box → instance mask AP |
| 0 | 47.8 | 48.1 | 28.6 | 13.1 |
| 10 | 48.0 | 46.2 | 28.0 | 11.2 |
| 20 | 48.6 | 46.6 | 28.5 | 11.7 |
| 30 | 47.1 | 45.8 | 27.4 | 11.7 |

Table 10: **Ablation study on the number of dataset-specific queries $\mathcal{Q}_d$.** Evaluated on a ResNet50 backbone. Without dataset-specific queries, the performance on COCO panoptic and Objects365 instance segmentation is much better. Using a large number of dataset-specific queries, *e.g.*, 30, hurts performance on many datasets, since it weakens knowledge sharing among datasets & tasks.

| | | w/ dataset-specific classification | w/o dataset-specific classification |
|---|---|---|---|
| ADE semantic | mIoU | 47.9 | 48.1 (+0.2) |
| COCO panoptic | PQ | 48.5 | 49.0 (+0.5) |
| ADE panoptic[†] | PQ | 28.7 | 29.8 (+1.1) |
| O365 instance[†] | AP | 12.9 | 14.3 (+1.4) |

Table 11: **Adding dataset-specific classification modules hurt performance on almost all datasets,** especially the weakly-supervised transfer results[†]. The dataset-specific classification modules include the classification embedding head and the learnable background classifier. Evaluated on a ResNet50 backbone, without other dataset-specific modules.

In Table 11, we compare using and not using dataset-specific classification modules (the classification embedding head and background classifier). Results indicate that removing dataset-specific classification improves performance on all datasets & tasks, especially the weakly-supervised tasks. We suspect it's because weakly-supervised tasks rely more on knowledge sharing.

**System-level comparison between with and without dataset-specific modules:** In the main paper, due to space limit, we only show results of adding all dataset-specific modules on a ResNet50 backbone. In Table 12, we show the same comparison on more backbones, *i.e.*, ResNet50, ViTDet-B, and ViTDet-L. The observation is consistent across all backbones: adding dataset-specific modules hurts performance on all datasets. The difference is more significant on backbones of smaller scales, and on COCO panoptic and Objects365 instance segmentation datasets.

# F  Comparison with related contemporary work

We discuss the differences between DaTaSeg and some contemporary related work as below.

**Comparison with Segment Anything [30]:** SAM [30] is a related work on multi-task universal segmentation model. There are several major differences between DaTaSeg and SAM. 1) The main use-case for SAM is prompt-based segmentation, i.e., the user inputs a point or box, and the model segments the object referred by the prompt. While we aim to segment all objects and stuff in a bottom-up way. 2) Besides, SAM built their own large-scale dataset with more than 1 billion masks (SA-1B) by designing a data engine consisting of model-assisted manual annotation and semi-automatic stage, which is very costly; while we only utilize multiple publicly available datasets and train a joint segmentation model to improve the performance. Hence, DaTaSeg and SAM's results are not comparable in terms of the training data. 3) Except for the explicitly trained text-to-mask task, SAM's outputs are class-agnostic binary masks (the SA-1B datasets are also class-agnostic), while our model outputs the class predictions together with the mask predictions.

**Comparison with X-Decoder [80]:** X-Decoder is a related work on multi-task segmentation. There are multiple differences between our work and X-Decoder, which makes it hard to have a strict comparison. We detail the differences below: 1) *Task and Focus*: The pre-training in X-Decoder includes panoptic segmentation, referring segmentation, and image-text pairs (image-text retrieval and image captioning), which has a focus on vision-language tasks. These tasks are very different from ours: we cotrain on three mainstream segmentation tasks (panoptic/semantic/instance) and exclusively focus on segmentation. 2) *Training paradigm*: We directly cotrain on multiple datasets using a shared set of parameters (single model), while the ADE and COCO results in X-Decoder Table 1 is task-specific transfer. That is, X-Decoder first pretrains on large-scale data and then fine-tunes on each target dataset using different sets of fine-tuned parameters. 3) *Training data*:

| Backbone | Model | Fully-Supervised | | Weakly-Supervised Transfer | |
|---|---|---|---|---|---|
| | | ADE semantic mIoU | COCO panoptic PQ | ADE semantic → panoptic PQ | O365 box → instance mask AP |
| ResNet50 | +D-S modules | 48.1 | 46.0 | 26.9 | 10.9 |
| | DaTaSeg | 48.1 (+0.0) | 49.0 (+3.0) | 29.8 (+2.9) | 14.3 (+3.4) |
| ViTDet-B | +D-S modules | 51.0 | 50.8 | 31.0 | 13.1 |
| | DaTaSeg | 51.4 (+0.4) | 52.8 (+2.0) | 32.9 (+1.9) | 16.1 (+3.0) |
| ViTDet-L | +D-S modules | 53.9 | 52.3 | 33.5 | 13.7 |
| | DaTaSeg | 54.0 (+0.1) | 53.5 (+1.2) | 33.4 (-0.1) | 16.4 (+2.7) |

Table 12: **Comparing DaTaSeg with the alternative architecture adding all dataset-specific modules on various backbones.** Results show that removing the dataset-specific modules improve performance on all datasets and all backbones. The gains are most significant on Objects365 instance and COCO panoptic segmentation (with dataset-specific modules, it cannot outperform the separately trained baseline on COCO panoptic).

X-Decoder pretrains on COCO panoptic and referring segmentation and 4M image-text pairs with a long schedule, and we quote from their paper: "all the pre-trained models are trained with 50 epochs of COCO data and roughly 45 epochs of 10 million image-text pairs". Afterwards, they fine-tune the model on COCO and ADE20k. By contrast, we only train on COCO panoptic, ADE20k semantic, and Objects365 v2 detection with a total of 1.8M of training images, which is much fewer than X-Decoder. We also do not train on large amounts of text data. 4) *Model architecture*: X-Decoder adopts Mask2Former. Our model architecture is different as explained in Sec. G. More importantly, we don't have an online text encoder in our model architecture.

# G   Architecture change from Mask2Former

Our network architecture is based on Mask2Former [8]. However, due to hardware difference (TPU vs. GPU), we need to make modifications to Mask2Former architecture to run on our hardware. Unlike GPUs, TPUs require a static computation graph with *fixed-shape* data (otherwise, it recompiles for each computation graph change, and causes significant slowdown). Therefore, we drop all the TPU-unfriendly operations in Mask2Former. In particular, we change the *multi-scale deformable attention transformer [79]* pixel decoder to a plain FPN [43]. The performance comparison is shown in Table 4(e) in the Mask2Former paper (51.9 PQ of deformable-attention vs. 50.7 PQ of FPN on COCO panoptic). Also, we do not use the *sampled point loss* [9] in either the matching loss or the training loss calculation, and use the vanilla mask loss. The performance comparison is shown in Table 5 in the Mask2Former paper (51.9 PQ of point loss vs. 50.3 PQ of mask loss). Besides, we *do not use varying input size* during evaluation and use a fixed input size (first resize then pad).

To improve the training speed, we *modify the deep supervision* in the masked attention transformer decoder: for every decoder layer, we use the same matching indices computed from the outputs of the last decoder layer, but use a different prediction head. We *do not use the mask predictions from the initial queries* as attention masks, since the predictions are not made based on the input image.

# H   Additional implementation details

In addition to the implementation details described in Sec. 4, we provide more datails here.

**Preprocessing.** In order to do 1:1 Hungarian matching, we preprocess the groundtruth into the same format as the predictions described in Sec. 3.2. In **panoptic segmentation**, we transfer the groundtruth into a set of binary masks with class labels. For "thing" categories, we group the pixels belonging to each instance into a separate groundtruth mask; for "stuff" categories, we group all pixels in each stuff category as the groundtruth mask. Similarly, in **semantic segmentation**, we group all pixels in each category as the groundtruth mask. For **instance segmentation**, since we are using bounding box weak supervision, we do not need this preprocessing step. For all mask proposal groundtruth, we downsample it by 4 times, to save memory cost during training.

**Training settings.** We randomly scale the input image in the range of [0.1, 2.0] and then pad or crop it to $1024 \times 1024$. For ADE20k dataset, since the image size is smaller than other datasets, we use a scaling range of [0.5, 2.0]. We use the AdamW optimizer [46] with a weight decay of 0.05. We

| | $\mu_{ce}$ | $\mu_{focal}$ | $\mu_{dice}$ | $\mu_{proj}$ | $\lambda_{ce}$ | $\lambda_{focal}$ | $\lambda_{dice}$ | $\lambda_{proj}$ |
|---|---|---|---|---|---|---|---|---|
| ADE semantic | 1 | 20 | 5 | 0 | 1 | 20 | 5 | 0 |
| COCO panoptic | 1 | 0 | 1 | 0 | 1 | 20 | 5 | 0 |
| O365 instance (box GT) | 1 | 0 | 0 | 0.5 | 1 | 0 | 0 | 2 |

Table 13: **The weights we use to compute the matching cost and total loss (Eqn. 4,5 in the main paper) for all training datasets.** $\mu$'s are for the matching cost and $\lambda$'s are for the total training loss.

clip the gradients with a max norm of 0.1. The weight for the background class is set to 0.05 in $\mathcal{L}_{ce}$. The matching cost and loss weight settings for Eqn. 4,5 in the main paper are shown in Table 13. We use a dataset sampling ratio of 1:4:4 for ADE semantic, COCO panoptic, and Objects365 detection. We adopt a different learning rate multiplier for each dataset: We multiply the learning rate on ADE semantic, COCO panoptic, and Objects365 detection by 3, 5, 2, respectively. Since Objects365 detection dataset has a large vocabulary with imbalanced distribution, we apply repeat factor sampling with a frequency threshold $t = 0.01$ [18]. On ResNet50 backbones, we use a batch size of 384 and train 500k iterations, with a learning rate of 3e-5. We adopt the step learning rate schedule: We multiply the learning rate by $0.1\times$ at the 0.9 and 0.95 fractions of the total training iterations. On the ViTDet-B backbones, we train 600k iterations with a learning rate of 6e-5. On ViTDet-L, we use a batch size of 256 and train 540.5k iterations with a learning rate of 4e-5. For ResNet50, we train on 64 TPU v4 chips; for ViTDet backbones, we train on 128 TPU v4 chips. All evaluations are conducted on 4 V100 GPUs with a batch size of 8.

**Postprocessing.** We first apply the `MERGE` operation described in Sec. 3.2. We follow the postprocessing operations in [8] with some modifications: In panoptic segmentation, we use a confidence threshold of 0.85 to filter mask proposals for COCO panoptic, and set the threshold to 0.8 for ADE panoptic. For panoptic and instance segmentation, we filter out final segments whose area is smaller than 4. For instance segmentation, we return a maximum of 100 instances per image with a score threshold of 0.0. For all tasks, the final scores are the production of classification scores and localization scores (obtained from binary mask classification).

# I    Limitations

Our proposed method is not without limitations. Since we are the first work to explore universal multi-dataset multi-task segmentation models, we do not introduce the complexity for an efficient framework. There are several ways to improve the model efficiency, *e.g.*, calculating the mask loss on sampled points only [8], while we do not deploy this in our framework, since it's not our key contribution. Moreover, as shown in the experiment section (Table 1), our framework is orthogonal to the detailed network architectures as long as the network is able to output our universal segmentation representation.

For weakly-supervised panoptic segmentation on ADE20k panoptic, we notice that sometimes multiple instances of the same category are not separated in the prediction. In the weakly-supervised instance segmentation results on Objects365, there are some visual artifacts in the mask prediction. We believe using certain data augmentation techniques, *e.g.*, MixUp [73] or Copy-Paste [15], may farther enhance knowledge transfer between datasets of different tasks, and thus may help mitigate this issue. We leave it for future work to improve on these limitations.

# J    Additional qualitative results

We notice sometimes our predictions do not match the groundtruth, but it does not necessarily mean the predictions are not good. Fig. 9 shows two examples on COCO panoptic. On the left column, DaTaSeg does a better job in segmenting 'sky' and 'tree-merged' than the groundtruth. The classification of 'pavement-merged' is also better than 'dirt-merged, which is probably due to *language ambiguity*: the British meaning of 'pavement' is sidewalk, while the annotator may be more used to the North American usage of 'pavement'. On the right column, DaTaSeg predicts 'window-other', while the groundtruth is 'tree-merged' for the scene through the window, and both make sense due to *label ambiguity*. In addition, DaTaSeg is able to predict the 'mirror-stuff', 'shelf', 'light', 'bottle', and more 'book's, which are missing from the groundtruth.

We show qualitative results for the direct transfer experiments in Fig. 10,11,12,13, on PASCAL Context 59 (PC-59), PASCAL Context 459 (PC-459), COCO semantic, and Cityscapes panoptic

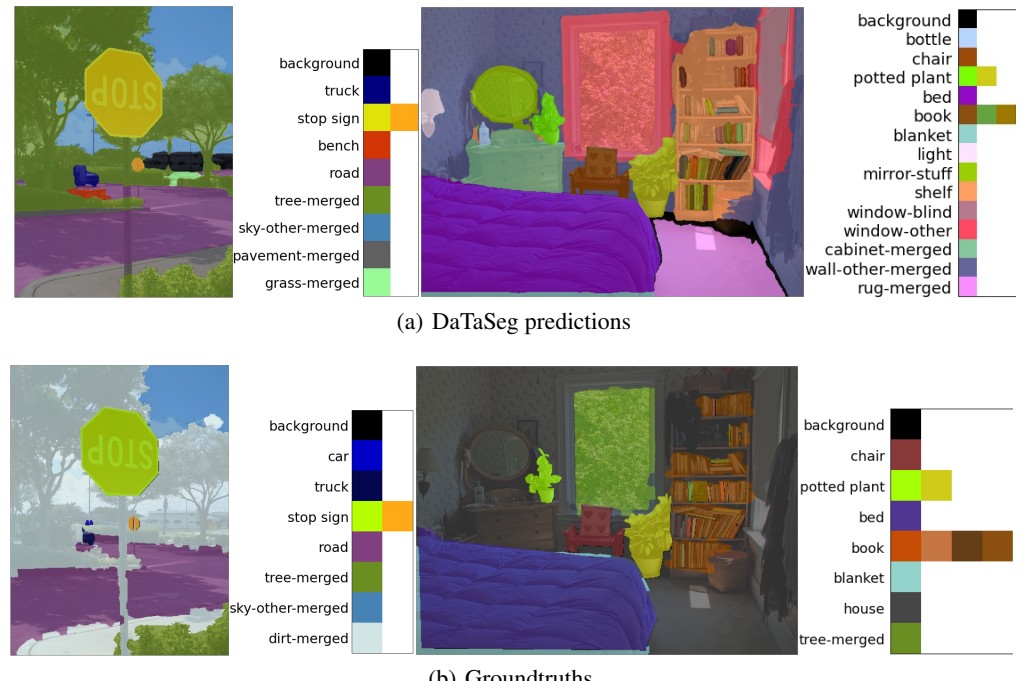

(a) DaTaSeg predictions

(b) Groundtruths

Figure 9: **Examples on COCO panoptic showing that sometimes DaTaSeg's predictions do not match the groundtruth, but it does not necessarily mean they are wrong.** On the left, DaTaSeg's prediction is 'pavement-merged', while the groundtruth is 'dirt-merged' (Probably because the British meaning of 'pavement' is sidewalk, while the annotator is more used to the North American meaning of 'pavement'). Ours also segments 'tree' and 'sky' better. On the right, the definition for the scene through the window is ambiguous: 'window-other' (ours) v.s. 'tree-merged' (GT). The groundtruth misses to label several objects, while DaTaSeg recognizes more objects, *e.g.*, 'mirror-stuff', 'light', 'shelf', 'bottle'. Legends for 'book' are truncated due to space limit.

datasets, respectively. These results serve as a supplementary material for Table 3 in the main paper. They show DaTaSeg directly transfer to other segmentation datasets unseen during training with high quality both in localization and classification. DaTaSeg performs well on large-vocabulary segmentation (PC-459), and handles hard cases well (thin structures, small objects, complicated scenes, occlusions, *etc.*).

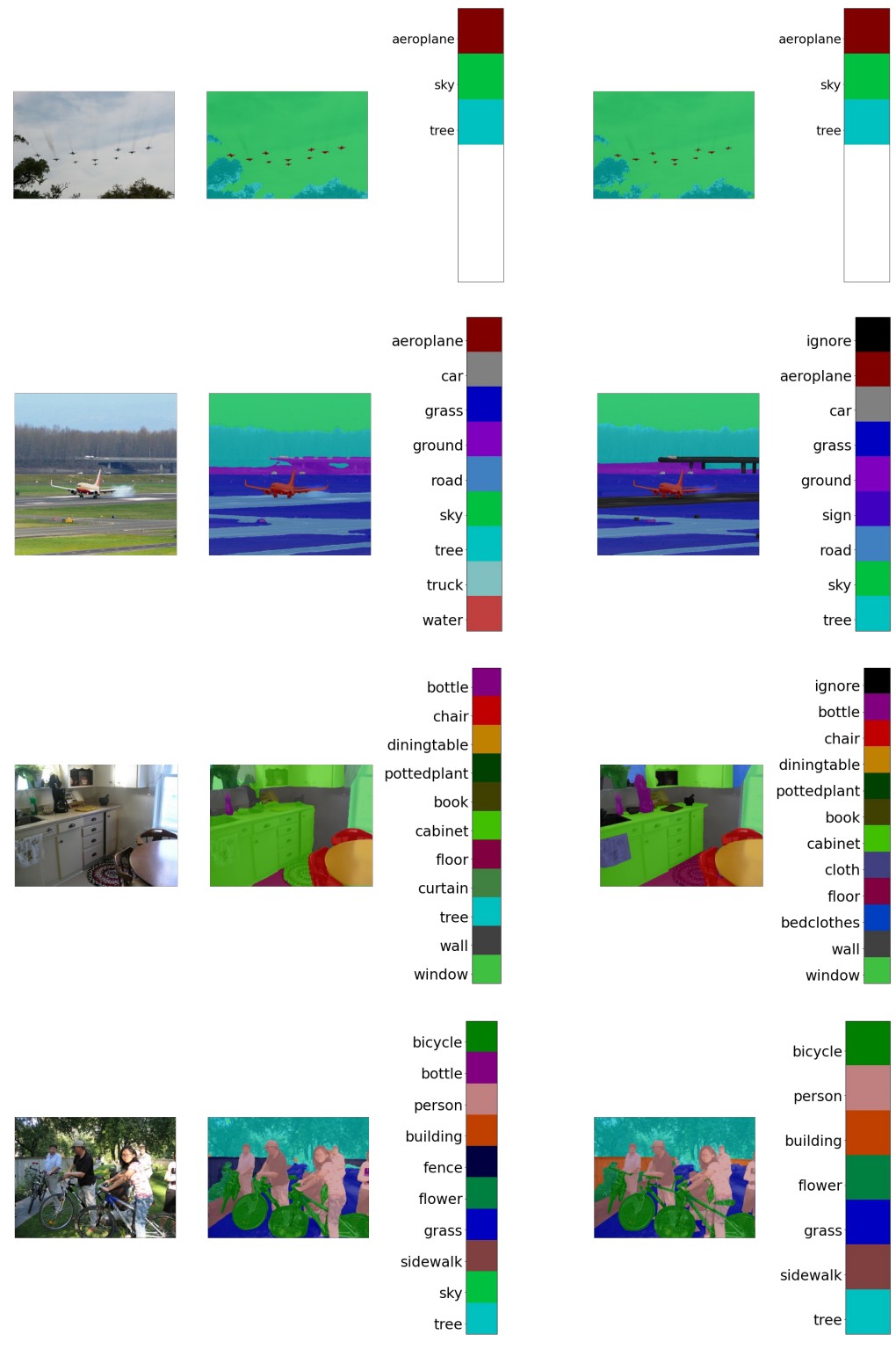

Figure 10: **Qualitative results of DaTaSeg directly transferring to PASCAL Context semantic dataset with 59 categories (PC-59).** The results demonstrate DaTaSeg has good generalization ability. The top row shows DaTaSeg is able to segment small objects (aeroplane), and the last row indicates DaTaSeg segments fine structures (bicycles) well. For each row, the left is the input image, the middle is our prediction, and the right is groundtruth. With a ViTDet-L backbone.

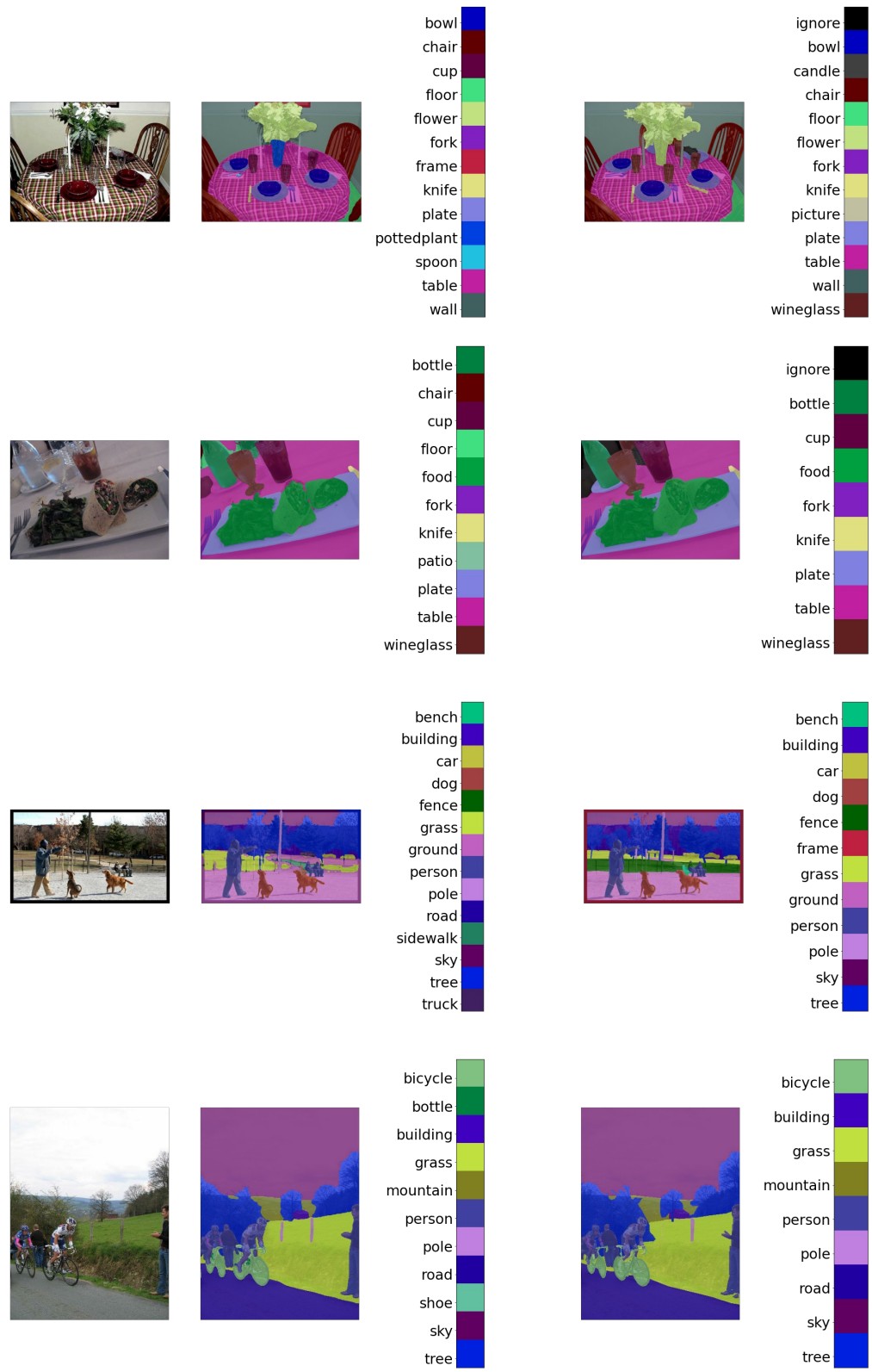

Figure 11: **Qualitative results of DaTaSeg directly transferring to PASCAL Context semantic dataset with 459 categories (PC-459).** The results demonstrate DaTaSeg has good generalization ability, and enables open-vocabulary segmentation. For each row, the left is the input image, the middle is our prediction, and the right is groundtruth. With a ViTDet-L backbone.

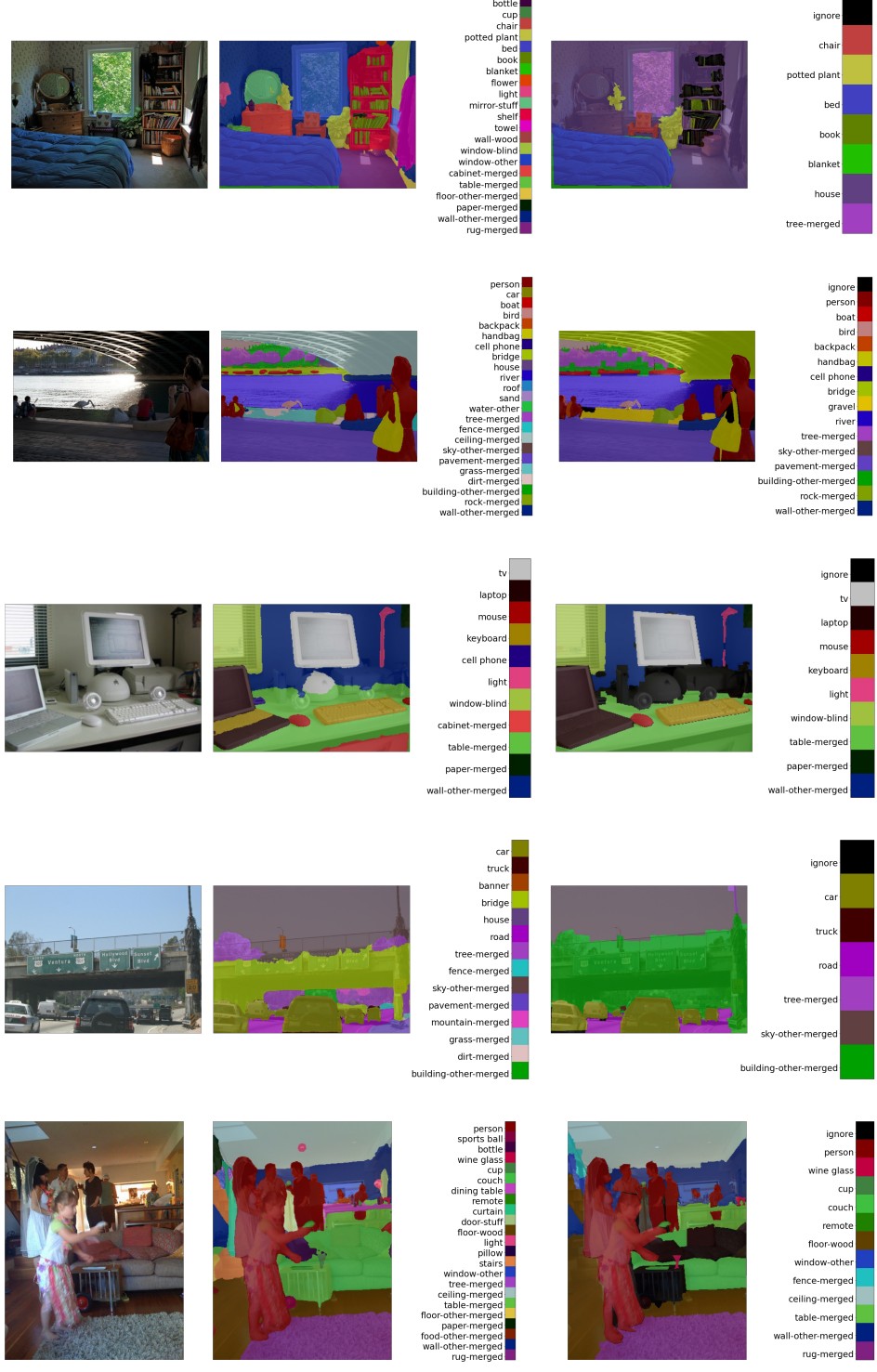

Figure 12: **Qualitative results on COCO semantic dataset.** DaTaSeg does high-quality semantic segmentation on COCO. Note that DaTaSeg trains on COCO panoptic. For each row, the left is the input image, the middle is our prediction, and the right is groundtruth. Our model is with a ViTDet-L backbone.

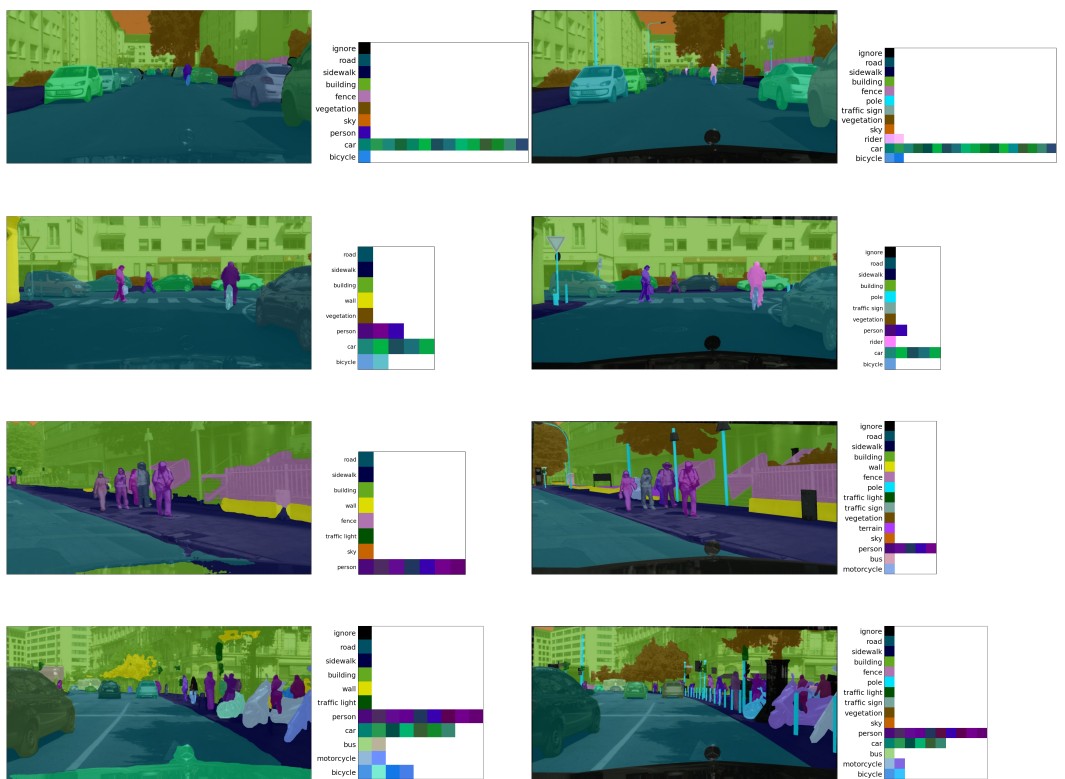

Figure 13: **Qualitative results of DaTaSeg directly transferring to Cityscapes panoptic dataset.** Cityscapes focuses on street view, which is different from all our training datasets. The results demonstrate good generalization ability. For each row, the left is our prediction and the right is groundtruth. Our model is with a ViTDet-L backbone.

