# OpenReview forum: "DaTaSeg: Taming a Universal Multi-Dataset Multi-Task Segmentation Model"
_NeurIPS.cc/2023/Conference — NeurIPS 2023 poster_

### Official Review · Reviewer_8rwi · 2023-07-03

**Soundness:** 2 fair
**Presentation:** 2 fair
**Contribution:** 2 fair
**Rating:** 5
**Confidence:** 5

**Summary:**

This paper proposes the universal model for image segmentation using multi-dataset multi-task training. By using universal segmentation representations at the entity (thing or stuff) level, the paper use merge operation for different segmentation task. Experiments show the effectiveness of the proposed method.

**Strengths:**

1, The paper is well-written and easy to understand.

2, The extensive experiments show the effectiveness of multi-dataset multi-task training.

**Weaknesses:**

1， Limited novelties. The main contribution of this paper is multi-dataset multi-task training. However, there is no obvious difference to OneFormer except for the multi-dataset training. To solve the label space conflict of multi-dataset training, the paper directly uses the language embeddings of the language model that has been widely explored.

2, Merging universal segmentation representations seems like following the fine-to-coarse pipeline. There is no obvious clue to prove whether this fine-to-coarse is better than the coarse-to-fine pipeline.

3, Weakly-supervised instance segmentation module also has limited novelties by just using the projection loss of BoxInst to add the larger dataset objects365.

**Questions:**

1, Could the author try the other sampling strategy in multi-dataset training? In the paper, it would lead to the imbalance of dataset sampling.

2, The performance improvement on COCO mainly comes from using the objects365 dataset. Direct pretraining object365 and then finetuning to COCO could perform better than the proposed method in the paper.

**Limitations:**

This paper cannot solve the annotation inconsistency of multi-dataset training. For example, the `window' defined in COCO and ADE20K are different where COCO stuff but ADE20K thing.

---

> ### Author Rebuttal · Authors · 2023-08-09
>
> We thank the reviewer for the helpful feedback.
>
> >**W1: Novelties and difference to OneFormer:**
>
> There are multiple significant differences between DaTaSeg and OneFormer. Though we discussed the most significant difference in L27 and L90, We detail more differences below:
>   1. Multi-dataset training is challenging (Reviewer EhPn) and using the same set of parameters (i.e., a single model) for multiple datasets saves computational resources (Reviewer 7Lho). OneFormer doesn't support multi-dataset training while DaTaSeg does. We emphasize that it's non-trivial to develop a single model for multi-dataset training, while attaining good performance on all datasets. The final simple yet effective design is in fact optimized from another more complicated design, as explained in supplementary Sec. D.
>   2. With our multi-dataset training, we 1) leverage knowledge from multiple datasets, containing different types of annotations, to improve segmentation performance on all datasets; 2) enable weakly-supervised instance and panoptic segmentation through knowledge transfer; 3) directly enable open-vocabulary segmentation. All these points are missing from OneFormer.
>   3. There are significant differences between DaTaSeg's and OneFormer's multi-task training strategy: we handle different segmentation tasks by post-processing (our MERGE operation), and OneFormer handles different segmentation tasks by defining a task prompt and using task-specific queries. This means OneFormer would have different model outputs for different tasks, while our model always produces the same, universal mask representation. Advantages of our framework include enhancing knowledge sharing among tasks, and potential better generalization to more future segmentation tasks.
>   4. Different supervision: OneFormer only uses fully-supervised training; while DaTaSeg explores weaker supervision. DaTaSeg achieves weakly-supervised instance segmentation using only bbox annotation.
>   5. Different network architectures: Except from the above differences, OneFormer first obtains a set of text queries using a text encoder and then applies a contrastive learning between the text queries and the object queries. DaTaSeg does not use such text queries.
>
> >**W2: Fine-to-coarse pipeline:**
>
> * The fine-to-coarse pipeline of merging universal segmentation representation is straightforward, simple, and effective.
> * Designing a coarse-to-fine pipeline is challenging, since we need to decompose semantic segmentation into meaningful instance segmentation masks.
> * Our fine-to-coarse pipeline is non-ambiguous, while ambiguity is hard to eliminate in a coarse-to-fine pipeline.
>
> >**W3: Novelties in weakly-supervised instance segmentation module:**
>
> * Our novelty lies in the proposed multi-dataset multi-task universal segmentation framework. We aim to design a simple, yet effective network architecture, and the projection loss from BoxInst fits our goal well.
> * We propose to use a universal segmentation representation (Sec. 3.1) with a fully-shared network architecture (Sec. 3.4), and apply different losses to different types of segmentation tasks.
> With the projection loss, we perform bounding-box-supervised instance segmentation *without any modification to the architecture*, which also performs well empirically.
> * To the best of our knowledge, we are the first to apply the projection loss in the multi-dataset multi-task segmentation setting.
>
> >**Q1: Sampling strategy in multi-dataset training:**
>
> * Our sampling strategy (L194-L199) avoids the imbalance of dataset sampling by specifying the per-dataset sampling ratio. The proportion of training samples coming from each dataset in the whole training process is determined by that sampling ratio.
> * In our main results, the sampling ratio is 1:4:4 for ADE:COCO:O365. We ablate the sampling ratio on a Resnet50 backbone and show the results in the table below.
>
> | Sampling ratio | ADE semantic | COCO panoptic | ADE panoptic | O365 instance |
> |:--------------:|:------------:|:-------------:|:------------:|:-------------:|
> |      1:4:4     |     48.1     |      49.0     |     29.8     |      14.3     |
> |      1:2:2     |     46.8     |      48.6     |     29.1     |      12.8     |
> |      1:1:1     |     45.3     |      48.0     |     28.4     |      13.7     |
>
> * Results show that our adopted sampling ratio is better than the other sampling ratios.
>
> >**Q2: "Improvement on COCO mainly comes from Objects365":**
>
> * In Table 2 of the submission, we have shown the results of training DaTaSeg on all combinations of the three datasets. With or without cotraining on Objects365, COCO performance is approximately the same. Hence, the performance improvement on COCO does not mainly come from using the Objects365 dataset.
> * We also experimented with training DaTaSeg from a checkpoint pretrained on O365 (3rd row in Table 2 in our submission), using ResNet50 backbone. The results are shown in the table below:
> | Pretrain data | ADE semantic | COCO panoptic | ADE panoptic | O365 instance |
> |--------------:|:------------:|:-------------:|:------------:|:-------------:|
> | IN+O365 |     47.6     |      48.7     |   28.8   |      13.6     |
> |      IN |     47.2     |      48.7     |     29.4     |      14.5     |
> * Results demonstrate that pretraining on O365 does not improve the performance for DaTaSeg. We hypothesize it's because weak bouding box pretraining doesn't help segmentation-only model, like DaTaSeg.
>
> >**Limitations: Annotation inconsistency of multi-dataset training:**
>
> * For stuff categories, we apply MERGE operation on mask predictions to obtain final predictions, during both training and inference. Therefore, our method can predict the window category as “stuff” category in COCO and “thing” category in ADE20k.
> This inconsistency is one motivation behind our design (L117-119): We first adopt a universal segmentation representation for different tasks, and then treat them differently in merging and postprocessing.

---

> > ### Comment · Reviewer_8rwi · 2023-08-18
> >
> > Thanks for the author's effort in rebuttal. It still cannot convince me very much.
> >
> > In Table 2, using both COCO panoptic and O365 datasets brings 18.3 mIoU on ADE semantic task. That means the annotation inconsistency still exists to some extent. It is similar to the setting about training the model on ADE semantic and O365 box and inference it on COCO Panoptic. Another reason is the image and task domain gap. That might be why the paper uses the ADE semantic instead of the ADE panoptic dataset.
> >
> > Also, merge operations can somewhat solve the annotation inconsistency. When doing inference, the user should select whether to adopt the merge operation based on different datasets.
> >
> > Anyway, I appreciate the difficulties of multi-dataset training and the heavy workload. Thus, I tend to vote for borderline acceptance. However, I cannot rate higher scores because the problem of the multi-dataset training for segmentation still needs to be solved.

---

> > > ### Author Response · Authors · 2023-08-19
> > >
> > > Dear Reviewer 8rwi,
> > >
> > > Thank you for recognizing the difficulties in our problem setting and our hard work!
> > >
> > > We carefully address your comments below.
> > >
> > > - **Table 2 performance:** When not training on ADE semantic or COCO panoptic, one primary reason for the lower performance is that the model only has limited knowledge about the categories in the untrained dataset. While it's interesting to transfer to untrained datasets, it's not the main problem that our submission is trying to address.
> > >
> > > - **Why we use ADE semantic instead of ADE panoptic:**
> > >   1. In our problem setting, we want to train on a suite of datasets of different segmentation tasks. ADE20K is one of the most widely-used benchmarks for semantic segmentation, so we choose to train on ADE20K semantic dataset. If we also train on ADE20K panoptic dataset, then the performance improvement on ADE20K semantic may come from ADE20K panoptic training (we note that semantic annotation is a subset of panoptic annotation which includes both semantic categories and instance identities), rather than from cotraining on other datasets -- which makes it harder to argue the benefits of multi-dataset multi-task training.
> > >   2. Besides, we are also curious about how well our model can perform on ADE20K panoptic without directly training on it, while only exploiting the weaker semantic annotations.
> > >
> > > - **Merge operation:** During inference time, the type of merge operation applied is decided by the desired segmentation task, as discussed in L151 of the paper. The user only needs to specify the segmentation task to perform.
> > >
> > > - **The problem of the multi-dataset training for segmentation is not solved:** Yes, we agree. This field is underexplored. Given the great benefits of multi-dataset multi-task segmentation models, we see our paper as one of the very first few works that take an initial step in this direction. We look forward to more future work to further improve the performance.

---

### Official Review · Reviewer_LhuR · 2023-07-05

**Soundness:** 3 good
**Presentation:** 3 good
**Contribution:** 3 good
**Rating:** 6
**Confidence:** 4

**Summary:**

This paper proposes DaTaSeg, a universal multi-dataset multi-task segmentation model. DaTaSeg uses a shared representation and different merge operations and post-processing for different tasks. Weak-supervision is employed for cheaper bounding box annotations and knowledge is sharing across different datasets with text embeddings from the same semantic embedding space and shared network parameters. A subset of the Objects365 validation set is annotated for instance segmentation. Experiments shows that DaTaSeg gets improved performance on dataset-specific models and enables weakly-supervised knowledge transfer. DaTaSeg also scales with the training dataset number and enables open-vocabulary segmentation.

**Strengths:**

1) This paper is easy to follow.
2) The method proposed in this paper will be useful in the future as it achieves the segmentation task under the multi-task multi-dataset setting.


**Weaknesses:**

The ablation study in the paper is somewhat not sufficient. The hyperparameters λ in equation (4) and μ in equation (5) are not provided and there needs to be experiments on the impact of hyperparameters.

**Questions:**

This paper proposes the multi-dataset multi-task segmentation model. The hyperparameters λ and μ in the equations (4) and (5) are crucial for the training. Could you provide the experiments on the impact of hyperparameters on the model performance?The authors do not discuss Limitations in the paper.

**Limitations:**

The authors do not discuss Limitations in the paper.

---

> ### Author Rebuttal · Authors · 2023-08-09
>
> We thank the reviewer for recognizing our contributions and the helpful feedback. We carefully address the comments below.
>
> >**Weaknesses and Questions: Hyperparameters and ablation study:**
>
> - Our settings of $\lambda$ and $\mu$ closely follow Maskformer [A] and Mask2former [B], and we add the weights for the bounding-box projection loss $L_{proj}$ for the O365 dataset.
> - We did an ablation study on the weights for the projection loss $L_{proj}$, $\lambda_{proj}$ and $\mu_{proj}$, using ResNet50 backbone, as shown in the table below:
>
> | $\lambda_{proj}$ | $\mu_{proj}$ | ADE semantic | COCO panoptic | ADE panoptic | O365 instance |
> |-----------------:|-------------:|:------------:|:-------------:|:------------:|:-------------:|
> |              5.0 |          1.0 |     45.5     |      48.3     |     28.4     |      12.5     |
> |              2.0 |          0.5 |     47.2     |      48.7     |     29.4     |      14.5     |
>
> - The results indicate that the 2nd setting (our final setting) has a better performance on all datasets.
> - Finding the optimal weights for multiple losses is an interesting but challenging problem, and we leave it for future work.
> - We will add the above table, and the tables in our rebuttal to reviewer t7xz and 8rwi to ablation studies in the revision.
>
> >**Limitations:**
>
> - We have a "Limitations" section in supplementary Sec. G.
>
>
> [A] Cheng, Bowen, Alex Schwing, and Alexander Kirillov. "Per-pixel classification is not all you need for semantic segmentation." NeurIPS 2021.
>
> [B] Cheng, Bowen, et al. "Masked-attention mask transformer for universal image segmentation." CVPR 2022.

---

### Official Review · Reviewer_t7xz · 2023-07-06

**Soundness:** 3 good
**Presentation:** 3 good
**Contribution:** 2 fair
**Rating:** 4
**Confidence:** 5

**Summary:**

[Tasks] This paper introduces DaTaSeg, a universal multi-dataset multi-task segmentation model that addresses the interconnections between panoptic, semantic, and instance segmentation tasks.

[Methods] DaTaSeg utilizes a shared representation, consisting of mask proposals with class predictions, across all tasks. To handle task discrepancies, the model employs distinct merge operations and post-processing techniques tailored to each task. Additionally, DaTaSeg leverages weak supervision, enabling cost-effective bounding box annotations to enhance the segmentation model. To facilitate knowledge sharing across datasets, DaTaSeg utilizes text embeddings from the same semantic embedding space as classifiers and shares all network parameters among datasets.

[Experiments] The model is trained on ADE semantic, COCO panoptic, and Objects365 detection datasets. DaTaSeg exhibits improved performance across all datasets, particularly for small-scale datasets, achieving a 54.0 mIoU on ADE semantic and a 53.5 PQ on COCO panoptic. Furthermore, DaTaSeg enables weakly-supervised knowledge transfer for ADE panoptic and Objects365 instance segmentation.

[Results] Experimental results indicate that DaTaSeg scales effectively with the number of training datasets and facilitates open-vocabulary segmentation through direct transfer.

[Dataset] Additionally, the authors have annotated an Objects365 instance segmentation set comprising 1,000 images, which will be released as a public benchmark.

**Strengths:**

1. The authors have conducted comprehensive experiments, showcasing the state-of-the-art performance achieved on multiple long-tailed recognition benchmarks. This highlights the robustness and effectiveness of their proposed method.

2. The paper is skillfully organized, making it easy to follow. The logical structure and clear presentation enhance the reader's understanding of the research.

3. Efficient Use of Supervision. While previous works have explored training universal models on multiple datasets and tasks, a notable strength of this paper is the effective utilization of weak bounding box supervision for segmentation. Compared to full mask annotations, weak bounding box supervision is a more cost-effective and practical solution. This approach makes the proposed method more accessible and applicable in real-world scenarios.

**Weaknesses:**

[Technical contributions on multi-task multi-task model.] One of the key contributions of this paper is the joint training of multiple datasets and multiple tasks within a unified framework. However, it should be noted that this aspect has already been explored by [1]. The referenced work employs a single set of parameters pre-trained for Semantic/Instance/Panoptic Segmentation, Referring Segmentation, Image Captioning, and Image-Text Retrieval tasks. Therefore, it might be necessary for the authors to clarify the differences between this work and X-Decoder.

[Technical contributions on text embedding classifier.] The utilization of a text embedding classifier has also been explored in previous works, such as [2]. In this work, an image encoder is trained to encode pixel embeddings, while CLIP text embeddings are employed as the per-pixel classifier. The key ideas are quite similar, although there are some differences on how to better leverage the text embeddings.

[Lack of comparisons with some published works and subpar performance compared to state-of-the-art methods.] The paper primarily compares the model's performance against methods designed or trained solely on a single task, referred to as "Separate" in the paper. However, it is worth noting that DaTaSeg benefits from training on a significantly larger sample size compared to these listed works. Consequently, the observed performance gains over the "Separate" models are not unexpected. Furthermore, there are published works, such as X-Decoder [1], that jointly train models on multiple datasets and tasks, yielding superior performance on many benchmarks compared to the reported results in this paper.

On ADE semantic (mIoU): X-Decoder achieves 58.1 compared to DaTaSeg's 54.0; on COCO Panoptic (PQ): X-Decoder outperforms DaTaSeg with 56.9 versus 53.5. In most benchmarks, DaTaSeg performs worse than X-Decoder.

[1] "Generalized decoding for pixel, image, and language." Zou, Xueyan, Zi-Yi Dou, Jianwei Yang, Zhe Gan, Linjie Li, Chunyuan Li, Xiyang Dai et al. In Proceedings of the IEEE/CVF Conference on Computer Vision and Pattern Recognition, pp. 15116-15127. 2023.
[2] "Language-driven Semantic Segmentation." Boyi Li and Kilian Q Weinberger and Serge Belongie and Vladlen Koltun and Rene Ranftl. International Conference on Learning Representations. 2022

**Questions:**

Most of my questions are listed in the Weaknesses section, with my main concerns focused on the technical contributions and performance comparisons with some published works. However, I have a couple of minor questions:

1. Will the code and model weights be made available to the public? It would be valuable to have access to these resources for further exploration and replication of the proposed method.

2. Given that this model can be trained with weakly-supervised tasks, I'm curious if further improvements in performance can be achieved by training on a larger quantity of weakly-supervised samples. Are there any trade-offs between the quality and quantity of the samples when it comes to enhancing model performance?

I believe addressing these questions would provide additional insights into the practicality and potential improvements of the proposed method.

**Limitations:**

Yes, the authors adequately addressed the limitations

---

> ### Author Rebuttal · Authors · 2023-08-09
>
> We thank the reviewer for the detailed and helpful comments, and for recognizing multiple strengths of our submission. We carefully address the comments and questions below.
>
> >**W1: Technical contributions on multi-task multi-task model:**
>
> - We train a single universal segmentation model on multiple segmentation tasks and multiple segmentation datasets, in one stage. As reviewer EhPn pointed out, we explore a very challenging task. However, the Table 1 results in the X-Decoder paper is **"task-specific transfer"**, which follows the “first pre-training then fine-tuning” paradigm to achieve the best performance on each dataset. It also requires separate fine-tuned model weights for each dataset.
> - The joint pre-training in X-Decoder only includes panoptic segmentation, referring segmentation, and image-text pairs, which has a focus on vision-language tasks. These tasks are very different from ours: we cotrain on three mainstream segmentation tasks (semantic/instance/panoptic) and exclusively focus on segmentation. This difference also leads to significantly different training data.
> - Besides, we novelly propose to include a simple weakly-supervised instance segmentation training using bounding-box supervision to increase the training data, since mask supervision is expensive and thus limited in scale.
> - We'll cite X-Decoder and include the comparison in the revision.
>
>
> >**W2: Technical contributions on text embedding classifier:**
>
> - We thank the reviewer for the question. We note that we have never claimed that it is our major novelty to use a text embedding classifier. Instead, our novelty lies in using the text embedding classifier for knowledge sharing among different datasets (L186-188). On the other hand, open-vocabulary segmentation, like [2], mainly uses it to support arbitrary text queries. In L190, we have acknowledged the same technique is used in open-vocabulary segmentation, and discussed the difference.
>
> >**W3: Comparisons with published works:**
>
> - We appreciate the comments, and agree that X-Decoder is a great work. However, there are multiple differences between our work and X-Decoder, which makes it hard to have a strict comparison. We detail the differences below:
>   1. **Training paradigm:** As we pointed out in W1, we directly cotrain on multiple datasets using a shared set of parameters (single model), while the ADE and COCO results in X-Decoder Table 1 is **task-specific transfer**. That is, X-Decoder first pretrains on large-scale data and then fine-tunes on each target dataset using different sets of fine-tuned parameters.
>   2. **Training data:** X-Decoder pretrains on COCO panoptic and referring segmentation and 4M image-text pairs with a long schedule, and we quote from their paper: *"all the pre-trained models are trained with 50 epochs of COCO data and roughly 45 epochs of 10 million image-text pairs"*. Afterwards, they fine-tune the model on COCO and ADE20k. By contrast, we only train on COCO panoptic, ADE20k semantic, and weakly-supervised Objects365 v2 detection with a total of 1.8M of training images, which is much fewer than X-Decoder.
>   3. **Model architecture:** X-Decoder adopts Mask2Former. Our model architecture is different as explained in supplementary Sec. E.
>
> - Finally, we note that X-Decoder is a concurrent work that was published in CVPR 2023, June 2023, while the NeurIPS submission deadline was in May 2023.
>
> - Therefore, we think comparing with the "separated" baselines under the same setting is reasonable to show our proposed multi-dataset multi-task training scheme improves segmentation performance.
>
> >**Q1: Open-sourcing:**
>
> - We thank the reviewer for the question. Indeed, we will release the code and model weights to the public, upon acceptance of the submission.
>
> >**Q2: Trade-offs between the quality and quantity of the training samples:**
>
> - Thanks for the great suggestion. In Table 2 of the submission, we show the results with and without cotraining with Objects365. The performance is generally better when cotraining with O365.
> - In addition, we experimented with cotraining on different portions of the O365 dataset: we cotrain DaTaSeg on COCO panoptic, ADE semantic, and 10%/25%/50%/100% of O365 training data, on the ResNet50 backbone. We show the results in the table below:
>
> | Ratio of O365 training data | ADE semantic | COCO panoptic | ADE panoptic | O365 instance |
> |--------------------:|:------------:|:-------------:|:------------:|:-------------:|
> |                 10% |     48.5     |      48.7     |     30.0     |      10.4     |
> |                 25% |     48.6     |      48.3     |     30.6     |      12.0     |
> |                 50% |     47.7     |      48.5     |     29.8     |      13.1     |
> |                100% |     47.2     |      48.7     |     29.4     |      14.5     |
>
> - The results show that when increasing the number of O365 weakly-supervised samples, O365 performance increases, COCO panoptic performance is not affected, and ADE20k performance slightly decreases. Overall, the gains are larger than the losses. We will include this interesting analysis in the revision.

---

> > ### Comment · Reviewer_t7xz · 2023-08-15
> >
> > I would like to extend my appreciation to the authors for their efforts in addressing my questions. While some of my concerns have indeed been adequately resolved, I must express that I still have reservations regarding the technical contributions and the comparisons made with certain published works.
> >
> > 1. The authors have argued that comparing the X-Decoder (XD) should be avoided due to its recent publication at CVPR 2023. I acknowledge this perspective, however, it's worth noting that ***X-Decoder was made available on ArXiv about 9 months*** ago and was accepted at CVPR around 5 months ago. Considering the dynamic nature of this field, I believe it is not unreasonable to consider a comparison.
> >
> > 2. In relation to the task variation, the authors have contended that XD's advantage largely stems from being trained on more data. It's important to note that the additional data largely comes from ***image-text paired data***, which ***is potentially easier to label than detection datasets***. I regard the ability to encompass a broader set of tasks as a strength rather than a weakness, particularly considering that the performance gains and the capability of supporting more tasks are attained through training on relatively *cheap* image-text pairs. \
> > Additionally, I would like to bring to the authors' attention the existence of OpenSeeD (accepted at ICCV2023) [2], which also harnesses detection data to enhance segmentation tasks. *Please note that I do not expect a comparison to [2] to be drawn. It's merely offered for your reference.*
> >
> > 3. The authors point out that the primary contributions stem from "include a simple weakly-supervised instance segmentation training using bounding-box supervision to increase the training data". Nonetheless, it's worth noting that weakly supervised instance segmentation is already a well-established task, and ***the Projection Loss employed in the paper to facilitate this approach is directly borrowed from [1]***. Given these factors, I find it challenging to regard this as a strong technical contribution. Furthermore, it can be argued which of the two is actually less resource-intensive: detection data or image-text data.
> >
> > Thank you once again for your time on addressing my questions!
> >
> > [1] Tian, Z., Shen, C., Wang, X. and Chen, H., 2021. Boxinst: High-performance instance segmentation with box annotations. In Proceedings of the IEEE/CVF Conference on Computer Vision and Pattern Recognition (pp. 5443-5452).
> >
> > [2] Zhang, H., Li, F., Zou, X., Liu, S., Li, C., Gao, J., Yang, J. and Zhang, L., 2023. A simple framework for open-vocabulary segmentation and detection. arXiv preprint arXiv:2303.08131.

---

> > > ### Author Response · Authors · 2023-08-19
> > >
> > > Dear Reviewer t7xz,
> > >
> > > We thank the reviewer for their review efforts and additional comments. Before we address the concerns, we would like to emphasize that as mentioned in the rebuttal, X-Decoder is a pioneering work and we are very willing to cite and compare it in our revision.
> > >
> > > Now, we carefully address the concerns below.
> > >
> > > >***Comparison with X-Decoder***:
> > >
> > > We simply added a note that X-Decoder is published at CVPR 2023. We did not argue that comparing with X-Decoder should be avoided --- We promised in our rebuttal W1: *"we'll cite X-Decoder and include the comparison in the revision"*.  Given the several key differences (rebuttal W1 and W3) between our work and X-Decoder, it's still hard to have a strict apple-to-apple comparison though. Below, we highlight the differences again:
> > >
> > >
> > >  1. **Task and Focus:** The pre-training in X-Decoder includes panoptic segmentation, referring segmentation, and image-text pairs (image-text retrieval and image captioning), which has a focus on vision-language tasks. These tasks are very different from ours: we cotrain on three mainstream segmentation tasks (panoptic/semantic/instance) and exclusively focus on segmentation.
> > >
> > >   2. **Training paradigm:** We directly cotrain on multiple datasets using a shared set of parameters (single model), while the ADE and COCO results in X-Decoder Table 1 is **task-specific transfer**. That is, X-Decoder first pretrains on large-scale data and then fine-tunes on each target dataset using different sets of fine-tuned parameters.
> > >
> > >   3. **Training data:** X-Decoder pretrains on COCO panoptic, referring segmentation and image-text pairs with a long schedule, and we quote from their paper: *"all the pre-trained models are trained with 50 epochs of COCO data and roughly 45 epochs of 10 million image-text pairs"*. Afterwards, they fine-tune the model on COCO and ADE20K separately. By contrast, we co-train on COCO panoptic, ADE20K semantic, and weakly-supervised Objects365 v2 detection with a total of 1.8M of training images. We do not train on large amounts of text data.
> > >
> > >   4. **Model architecture:** X-Decoder adopts Mask2Former. Our model architecture is different as explained in supplementary Sec. E. More importantly, we don't have an ***online text encoder*** in our model architecture.
> > >
> > > Consequently, having a strict apple-to-apple comparison will require a careful alignment of each setting, which is beyond the scope of this work and rebuttal.
> > >
> > > >***Image-text vs. detection data:***
> > >
> > > We agree with the reviewer that *the ability to encompass a broader set of tasks as a strength rather than a weakness*, and we never claim that it is a weakness. This is also one of our motivations to include various types of segmentation annotations from multiple datasets for training a single model.
> > >
> > > We emphasize that both image-text paired data and detection data are valuable training data. However, it is beyond the scope of this work and rebuttal to compare which one is better.
> > >
> > > Finally, we thank the reviewer for bringing OpenSeeD to our attention. We are also happy to cite OpenSeeD in our final revision.
> > >
> > > >***Projection loss:***
> > >
> > > We thank the reviewer for the comment. We emphasize that we never claim that it is our *primary contribution* to use the projection loss. We have carefully phrased our contributions in the draft and rebuttal. Our primary novelty is that we **propose a single unified framework for multi-dataset multi-task segmentation**. We list other contributions below:
> > >
> > >   1. With our multi-dataset multi-task training, we 1) leverage knowledge from multiple datasets, containing different types of annotations, to improve segmentation performance on all datasets, especially smaller-scale datasets, such as ADE20K; 2) enable weakly-supervised instance and panoptic segmentation through knowledge transfer; 3) directly enable open-vocabulary segmentation.
> > >
> > >   2. We propose to use a universal segmentation representation (Sec. 3.1) with a fully-shared network architecture (Sec. 3.4), and apply different losses to different types of segmentation tasks. With the projection loss, we perform bounding-box-supervised instance segmentation *without any modification to the architecture*, which also performs well in the co-training empirically.

---

### Official Review · Reviewer_7Lho · 2023-07-07

**Soundness:** 4 excellent
**Presentation:** 3 good
**Contribution:** 4 excellent
**Rating:** 7
**Confidence:** 4

**Summary:**

This paper proposes DaTaSeg, a general multi-dataset multi-task segmentation model. It utilizes shared representations and different pooling operations to perform panoramic, semantic and instance segmentation tasks. DaTaSeg benefits from weak supervision and knowledge transfer across datasets. It outperforms separate training on all datasets (especially smaller ones) and enables weakly supervised segmentation. The model also transfers well to unseen datasets and supports open vocabulary segmentation.

**Strengths:**

The paper addresses the challenge of training a single model on multiple segmentation tasks and datasets by proposing DaTaSeg, which has the potential to save computational resources and streamline the development of segmentation models. It can also benefit from weak supervision by incorporating cheaper bounding box annotations.
The proposed DaTaSeg shows promising results in transferring to unseen datasets and enabling open-vocabulary segmentation.
The authors annotate a subset of the Objects365 dataset and plan to release it as a public benchmark for instance segmentation. This contributes to the research community by providing a standardized evaluation dataset.

**Weaknesses:**

The Segment Anything model (SAM) released in April  2023 should be included for comparison as it's also a universal segmentation model which has zero/few-shot capability.

**Questions:**

May need proofreading:

Line 35, "which are" -> "which is"
Line 38, "which map" -> "which maps"

**Limitations:**

According to the F section in Supplementary Material, the computational cost is quite high which may affect the reproducibility. Consider releasing the pre-trained models in various sizes.

---

> ### Author Rebuttal · Authors · 2023-08-09
>
> We thank the reviewer for recognizing our contributions and the helpful feedback. We carefully address the comments below.
>
> >**Weaknesses: Comparison with SAM:**
>
> We thank the reviewer for the suggestion. We are happy to include a comparison with the Segment Anything paper in the revision. We note that SAM is a great paper, but there are several major differences between DaTaSeg and SAM.
>   1. The main use-case for SAM is prompt-based segmentation, i.e., the user inputs a point or box, and the model segments the object referred by the prompt. While we aim to segment all objects and stuff in a bottom-up way. The only instance segmentation quantitative results reported in the SAM paper (Table 5) requires taking boxes from an external trained detector ViTDet.
>   2. Besides, SAM built their own large-scale dataset with more than 1 billion masks (SA-1B) by designing a data engine consisting of model-assisted manual annotation and semi-automatic stage, which is very costly; while we only utilize multiple publicly available datasets and train a joint segmentation model to improve the performance. Hence, our DaTaSeg and SAM are not comparable in terms of the training data.
>   3. Third, except for the explicitly trained text-to-mask task, SAM's outputs are class-agnostic binary masks (the SA-1B datasets are also class-agnostic), while our model outputs the class predictions together with the mask predictions.
>
>
> >**Questions: Proofreading:**
>
> Thanks! We will fix these typos and do a thorough proofreading.
>
>
> >**Limitations: Computational cost and open-sourcing:**
>
> - We thank the reviewer for the suggestion. Indeed, we plan to open-source the code and release the models in various sizes, upon acceptance of the paper.
> - We use a longer training schedule because we are cotraining on the much larger Objects365 dataset (1,662,292 training images), which is 14 times larger than COCO. In order to balance the samples from different datasets, we train longer on COCO and ADE.
> - We have some explanation about the computational efficiency in supplementary Sec. G, including several techniques to improve it. We leave it for future work to further improve the efficiency.

---

### Official Review · Reviewer_EhPn · 2023-07-08

**Soundness:** 3 good
**Presentation:** 3 good
**Contribution:** 4 excellent
**Rating:** 6
**Confidence:** 5

**Summary:**

This paper introduces DaTaSeg, a universal multi-dataset multi-task segmentation model. It uses a shared representation for panoptic, semantic, and instance segmentation tasks, with different techniques to address task differences. Weak supervision and knowledge sharing are employed. DaTaSeg improves performance on various datasets, especially small-scale ones. It enables knowledge transfer and open-vocabulary segmentation. An Objects365 instance segmentation set of 1,000 images will be released as a benchmark.

**Strengths:**

1. It is noticed that training segmentation models on multiple datasets to obtain better results is difficult. It is more challenging to train segmentation models on multiple datasets for multiple tasks (instance segmentation, semantic segmentation, weakly supervised instance segmentation. The performance of DaTaSeg on COCO and ADE shows the effectiveness of the proposed approach
2. The paper leverages box-level supervision to improve the segmentation performance.


**Weaknesses:**

1. [minor] The definition of mask proposal is commonly used in instance segmentation [1]. However, the description uses too many math terms to describe a simple concept that can be demonstrated by natural language, figures, and more compact math. This is not good for readers to understand.

2. [major] The results of weakly supervised instance segmentation look not promising. In the previous state-of-the-art method, weakly supervised instance segmentation methods achieves 70-90% performance (mAP) as what their fully supervised versions do [2,3]. However, in the 2, the instance segmentation of Object365 looks not promising, only 10ish%. What is the upper-bound performance of O365 instance segmentation?

3. [major] The approach introduces the unified mask representation without comparison or discussion with other mask representations. It is well-known that training multi-dataset or multi-tasks is not trivial, and it is better to show more training details and empirical design considerations in the paper.



[1] run-length encoding: https://github.com/cocodataset/cocoapi/blob/master/PythonAPI/pycocotools/coco.py#L265
[2] Lan, Shiyi, et al. "Vision transformers are good mask auto-labelers." Proceedings of the IEEE/CVF Conference on Computer Vision and Pattern Recognition. 2023.
[3] Li, Wentong, et al. "Box2Mask: Box-supervised Instance Segmentation via Level-set Evolution." arXiv preprint arXiv:2212.01579 (2022).

**Questions:**

None

---

> ### Author Rebuttal · Authors · 2023-08-09
>
> We thank the reviewer for recognizing our contributions and the helpful feedback. We carefully address the comments below.
>
> >**W1: Mask proposal:**
>
> - We agree with the reviewer that the mask proposal is commonly used in instance segmentation. However, mask proposals are used differently in different segmentation tasks (e.g., one mask proposal corresponds to one instance mask in instance segmentation, but it may correspond to one amorphous stuff region in semantic or panoptic segmentation). Therefore, Sec. 3.2 is mainly about how we novelly apply *different* merge operations for *different* segmentation tasks. We briefly introduce the mask proposal representation in L121 in a math format, in order to use it in Eq. 1 and 2, which are our novelly proposed MERGE operation. We thank the reviewer for the suggestion and will improve the presentation.
>
> >**W2: Weakly supervised instance segmentation:**
>
> We thank the reviewer for the question, and address it below.
>
> - First, the reason why the scores on O365 are generally low in absolute number is because: O365 has 365 categories with imbalanced distribution [A]; hence, it is more challenging than detection datasets on common categories, e.g., COCO.
> - Second, we also evaluate DaTaSeg's class-agnostic mask average recall (AR@100) on O365. DaTaSeg achieves **32.8** and **34.5** AR@100 using ResNet50 and ViTDet-B backbones, respectively, which is not low.
> - Third, since there is no O365 instance segmentation training set available (it is one of our contributions to annotate 1000 O365 images with instance segmentation annotation, used as a validation set for weakly supervised instance segmentation), we do not have a fully-supervised upper bound. That being said, we have also tried our best to find an **approximate** upper-bound performance: in Detic [A], their fully-supervised detector on Objects365 is 31.2 box AP with a Swin-B backbone, which additionally uses several techniques (e.g., Federated loss  [B] and repeated factor sampling [C]) to address the category imbalance issue. Since mask AP is generally several points lower than box AP (e.g., in Detectron2 Mask R-CNN benchmark, COCO box AP can be 5-6 points higher than mask AP), our performance on the comparable ViTDet-B backbone of 16.1 mask AP is reasonable.
> - Finally, we thank the reviewer for the pointers. We will include [2,3] in the related work section. Ours and [2,3] focus on different aspects. [2] first generates high-quality pseudo masks using groundtruth bounding boxes, and then trains an instance segmentation model. Its pipeline is more complicated than ours. [3] proposes several complex modules specifically for box-supervised instance segmentation, while our framework is simpler and tackles multiple segmentation tasks simultaneously.
>
> >**W3: Comparison with other mask representations:**
>
> - For the mask representation, we explain our motivation in L110-113.
> - Another commonly used mask representation is first getting the bounding box prediction, and then predicting the mask within the predicted bounding-box region, as in Mask R-CNN. However, this representation is not suitable for stuff categories, e.g., it is unnatural to get a single bounding box for multiple grass fields. As a result, some methods such as [D] additionally add a semantic segmentation branch for stuff categories. By contrast, in the mask representation stage, we did not treat thing and stuff categories differently. We explain the reason in L116 — different datasets have different definitions for things and stuff, e.g., the 'table' category; so treating thing and stuff mask representation separately is unsuitable. Instead, we apply different merge operations and postprocessing in the later stage.
> - We agree with the reviewer that *”it is better to show more training details”*. We have already provided more empirical design considerations in the supplementary Sec. D, E, and more implementation details in supplementary Sec. F.
>
>
> [A] Zhou, Xingyi, et al. "Detecting twenty-thousand classes using image-level supervision." ECCV 2022.
>
> [B] Zhou, Xingyi, and Philipp Krähenbühl. "Joint COCO and LVIS workshop at ECCV 2020: LVIS challenge track technical report: CenterNet2." 2020.
>
> [C] Gupta, Agrim, Piotr Dollar, and Ross Girshick. "Lvis: A dataset for large vocabulary instance segmentation." CVPR 2019.
>
> [D] Kirillov, Alexander, et al. "Panoptic feature pyramid networks." CVPR 2019.

---

### Decision · Program_Chairs · 2023-09-21

**Decision:**

Accept (poster)

**Comment:**

This paper proposes an approach to unify different segmentation datasets and different tasks like panoptic, semantic and instance segmentation tasks.  This is a challenging setting and the proposed DaTaSeg demonstrate promising results.  Concerns were raised regarding comparisons with alternative methods, implementation details and discussion with some up-to-date references.

The authors made good effort in  the rebuttal and clarified some technical details and added more experimental results. The AC recommends acceptance, and encourages the authors to revise the paper accordingly.